# The mTORC2 subunit RICTOR drives breast cancer progression by promoting ganglioside biosynthesis through transcriptional and epigenetic mechanisms

Mohammad Nafees Ansari[1,2�], Somesh K. Jha[3�], Ali Khan[1�], Kajal Rajput[1], Nishant Pandey[3], Dolly Jain[3], Rajeshwari Tripathi[4], Nihal Medatwal[3], Pankaj Sharma[1], Sudeshna Datta[2], Animesh Kar[3], Trishna Pani[1], Sk Asif Ali[2], Kaushavi Cholke[1], Kajal Rana[3], Valiya P. Snijesh[5,6], Geetashree Mukherjee[7], Suryanarayana V. S. Deo[8], Soumen Basak[2], Ashutosh Mishra[8], Jyothi S. Prabhu[5], Arnab Mukhopadhyay[2], Avinash Bajaj[3]*, Ujjaini Dasgupta [iD][1,4]*

1 Amity Institute of Integrative Sciences and Health, Amity University Haryana, Gurgaon, Haryana, India, 2 National Institute of Immunology, New Delhi, India, 3 Regional Centre for Biotechnology, NCR Biotech Science Cluster, Faridabad, Haryana, India, 4 Koita Centre for Digital Health-Ashoka, Trivedi School of Biosciences, Ashoka University, Sonipat, India, 5 Division of Molecular Medicine, St. Johns Research Institute, St Johns National Academy of Health Sciences, Bangalore, Karnataka, India, 6 Centre for Doctoral Studies, Manipal Academy of Higher Education (MAHE), Manipal, Karnataka, India, 7 Tata Medical Center, Kolkata, West Bengal, India, 8 Department of Surgical Oncology, BRA-Institute Rotary Cancer Hospital, All India Institute of Medical Sciences, New Delhi, India

she These authors contributed equally to this work.
* bajaj@rcb.res.in (AB); ujjaini.dasgupta@ashoka.edu.in (UD)

## Abstract

Sphingolipid and ganglioside metabolic pathways are crucial components of cell signaling, having established roles in cancer cell proliferation, invasion, and migration. However, regulatory mechanisms controlling sphingolipid and ganglioside biosynthesis in mammalian cells are less known. Here, we show that RICTOR, the regulatory subunit of mTORC2, regulates the synthesis of sphingolipids and gangliosides in human luminal breast cancer-specific MCF-7 and BT-474 cells through transcriptional and epigenetic mechanisms. We observe that RICTOR regulates glucosylceramide levels by modulating the expression of UDP-Glucose Ceramide Glucosyl transferase (UGCG). We identify Zinc Finger protein X-linked (ZFX) as a RICTOR-responsive transcription factor whose recruitment to the *UGCG* promoter is regulated by DNA methyltransferase 1 and histone demethylase (KDM5A), which are known AKT substrates. We further demonstrate that RICTOR regulates the synthesis of GD3 gangliosides through ZFX and UGCG, and triggers the activation of the EGFR signaling pathway, thereby promoting tumor growth. In line with our findings in human cell culture and mouse models, we observe an elevated expression of RICTOR, ZFX, and UGCG in Indian luminal breast cancer tissues and in TCGA and METABRIC datasets. Together, we establish a key regulatory circuit,

**Data availability statement:** All relevant data are presented within the figures and supplemental figures of the paper, and numerical data are provided in the Supporting S1 Data–S6 Data for each main figure and in S1–S6 Datasets for the supplemental figures, with individual tabs for each supplementary figure.

**Funding:** This project was supported by the following grants: UD: BT/PR40413/BRB/10/1922/2020, BT/PR19634/BIC/101/488/2016, BT/PR48044/MED/30/2443/2023 Department of Biotechnology, Government of India. https://dbtindia.gov.in/ UD: SPF/2023/000075, Anusandhan National Research Foundation, India, https://www.anrfonline.in/ANRF/HomePage AB: BT/PR40413/BRB/10/1922/2020, BT/PR48044/MED/30/2443/2023 Department of Biotechnology, Government of India. https://dbtindia.gov.in/ AM: STR/2019/000064, Anusandhan National Research Foundation, India, https://www.anrfonline.in/ANRF/HomePage AM: BT/HRD/NBA/38/04/2016, Department of Biotechnology, Government of India https://dbtindia.gov.in/. The funders did not play any role in the study design, data collection and analysis, decision to publish, or preparation of the manuscript.

**Competing interests:** The authors have declared that no competing interests exist.

**Abbreviations:** AIIMS, All India Institute of Medical Sciences; ATCC, American Type Culture Collection; CAFs, cancer-associated fibroblasts; CPCSEA, Committee for Purpose of Control and Supervision of Experiments on Animals; EGFR, epidermal growth factor receptor; EMSA, Electrophoretic Mobility Shift Assay; GEMs, glycolipid-enriched microdomains; GEO, Gene Expression Omnibus; IAEC, Institutional Animal Ethical Committee; MRM, multiple reaction monitoring; RGCIRC, Rajiv Gandhi Cancer Institute and Research Center; RTKs, Receptor-Tyrosine Kinases; TCGA, The Cancer Genome Atlas; UGCG, UDP-Glucose Ceramide Glucosyltransferase; UMAP, Uniform Manifold Approximation and Projection.

RICTOR-AKT-ZFX-UGCG-Ganglioside-EGFR-AKT, and elucidate its contribution to breast cancer progression.

## Introduction

Sphingolipids and gangliosides are metabolically interconnected structural and signaling components of the cell membranes, and deregulation in their metabolism is linked to key human diseases, including cancer [1,2]. Ceramides are the central hub of the sphingolipid metabolic pathway, and modifications at their 3′-hydroxyl terminal lead to structurally diverse classes of sphingolipids like glucosylceramides, sphingomyelins, and ceramide-1-phosphates with a distinct role in different facets of tumourigenesis (Fig 1A) [3]. Synthesis of glucosylceramides from ceramides is catalyzed by UDP-Glucose Ceramide Glucosyltransferase (UGCG), and ceramide-glucosylceramide rheostat connecting sphingolipid and ganglioside metabolism plays a crucial role in tumor progression, drug resistance, and chemotherapeutic response (Fig 1A) [1]. Gangliosides are sialic acid-containing glycosphingolipids residing in glycolipid-enriched microdomains (GEMs) of the plasma membrane [4]. Gangliosides, as residents of GEMs, can induce activation/deactivation of Receptor-Tyrosine Kinases (RTKs), thereby regulating the downstream signaling processes [5]. Therefore, understanding the mechanisms that regulate the ceramide-glucosylceramide rheostat in cancer cells, connecting sphingolipid and ganglioside pathways, and its impact on RTK signaling may potentially lead to the identification of new therapeutic targets.

PI3K/AKT/mTOR is the key downstream pathway of most RTKs [6] and is hyperactivated in ~60% of breast cancer patients due to increased expression of growth factors and their receptors or genetic alterations in *PIK3CA, AKT*, and *PTEN* [7]. The mammalian target of Rapamycin (mTOR), an intracellular serine/threonine kinase of the PI3K pathway, exists in two different complexes, mTORC1 and mTORC2. Rapamycin-Insensitive Companion of mTOR (RICTOR) is the key regulatory subunit of mTORC2 that can directly phosphorylate AKT at Ser$^{473}$ [8]. Apart from AKT activation, mTORC2 also regulates tumourigenesis through activation of other substrates like AGC kinases, serum- and glucocorticoid-induced protein kinase 1 (SGK), and protein kinase C (PKC) [9–12]. Amplification of *RICTOR* or elevated RICTOR protein expression is positively correlated with poor overall survival of cancer patients [13–17]. Meta-analysis of cancer genomics datasets shows the co-occurrence of RTK alterations with RICTOR overexpression in different cancer types [18]. Therefore, there is a need to unravel the role of RICTOR (mTORC2)-mediated regulatory pathways in tumourigenesis.

Metabolic reprogramming is one of the hallmarks of tumourigenesis, and mTORC2 has emerged as a key link between RTK signaling and cancer metabolic reprogramming [19]. mTORC2 can regulate glycolytic metabolism in cancer cells through AKT phosphorylation, regulating c-Myc expression, or through transcription factor, FoxO [19]. mTORC2 also regulates lipogenesis, lipid homeostasis, and adipogenesis in noncancer cells; however, mTORC2-mediated regulation of sphingolipids

Luminal tumours have a deregulated sphingolipid and ganglioside profile and high RICTOR expression.

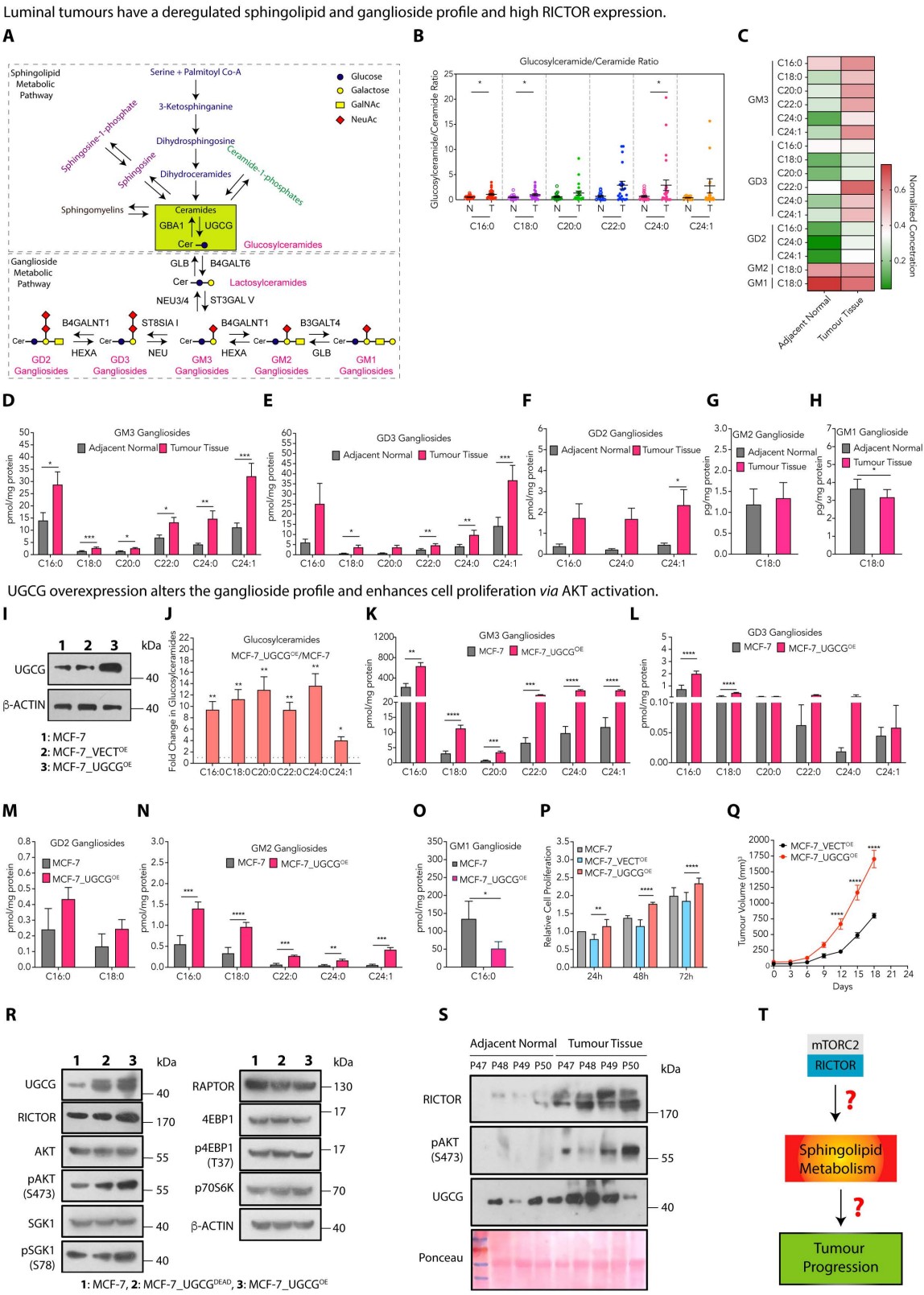

UGCG overexpression alters the ganglioside profile and enhances cell proliferation *via* AKT activation.

**Fig 1. Luminal tumors have a deregulated sphingolipid and ganglioside profile and high RICTOR expression.** (**A**) A schematic presentation showing the ceramide-glucosylceramide rheostat connecting the sphingolipid and ganglioside metabolic pathways. (**B**) The glucosylceramide to ceramide ratio (mean ± SEM, $n = 27$) for luminal tumor tissues (labeled as T) and adjacent normal tissues (labeled as N) indicates that the balance is shifted towards glucosylceramides. (**C**) Heat map showing the levels of different ganglioside species in luminal tumor tissues and adjacent normal tissues. (**D–H**) Absolute quantification (mean ± SEM, $n = 5$) of GM3 (D), GD3 (E), GD2 (F), GM2 (G), and GM1 (H) gangliosides in luminal tumor tissues and adjacent normal tissues shows an increase in GM3 and GD3 gangliosides and a decrease in GM1 gangliosides in luminal tumor tissues. (**I**) Immunoblot confirming an increase in UGCG expression in MCF-7_UGCG[OE] cells. (**J**) Absolute quantification of glucosylceramides (mean ± SEM, $n = 5$) confirms an increase in MCF-7_UGCG[OE] cells over MCF-7 cells. (**K–O**) Absolute quantification (mean ± SEM, $n = 5$) of GM3 (K), GD3 (L), GD2 (M), GM2 (N), and GM1 (O) ganglioside species shows an increase of GM3, GD3, and GM2 gangliosides and a decrease of GM1 gangliosides in MCF-7_UGCG[OE] cells compared to MCF-7 cells. (**P**) Cell proliferation (mean ± SEM, $n = 5$) assay demonstrates an increase in the proliferation of MCF-7_UGCG[OE] cells over MCF-7_VECT[OE] cells. (**Q**) Tumor growth kinetics reveal enhanced growth of MCF-7_UGCG[OE] tumors compared to MCF-7_VECT[OE] tumors (mean ± SEM, $n = 5–6$). (**R**) Immunoblots show the expression of RICTOR, RAPTOR, AKT, pAKT[Ser473], SGK1, pSGK1[Ser78], 4EBP1, p4EBP1[Thr37], and p70S6K in MCF-7_UGCG[OE] cells in comparison to MCF-7_UGCG[DEAD] cells. (**S**) Immunoblots for RICTOR, pAKT[Ser473], and UGCG in tumor tissues from luminal cancer patients show higher expression than adjacent normal tissues. (**T**) A schematic diagram showing the questions to be answered to understand the mTORC2-mediated regulation of the sphingolipid metabolic pathway and its role in tumor progression. Data among groups were analyzed using a paired Student $t$ test (for patient data), One-way ANOVA among multiple groups or Two-way ANOVA in time-dependent studies. *P*-value: *$p < 0.05$, **$p < 0.01$, ***$p < 0.0005$, ****$p < 0.0001$. Numerical data can be found in S1 Data.

and gangliosides in cancer cells is not well understood [20]. Yeast studies have shown that TORC2-dependent protein kinase, Ypk1, activates phosphorylation of subunits of the ceramide synthase complex, leading to an increased synthesis of sphingolipids [21]. It is also known that membrane stress in *S. cerevisiae* causes a redistribution of Slm proteins in the plasma membrane, leading to their association with the TORC2 complex and further activation of sphingolipid synthesis [22]. Recent studies have shown that mTORC2 may promote tumourigenesis in hepatocarcinoma by promoting de novo fatty acid and lipid synthesis [23]. However, mTORC2-mediated regulation of sphingolipid and ganglioside metabolism through transcriptional and epigenetic regulatory mechanisms and the impact of the network surrounding this metabolic hub on tumourigenesis is not known. Here, we delineate mTORC2-mediated regulation of ceramide-glucosylceramide rheostat and its role in breast cancer progression.

Luminal (A and B) is the most common breast cancer subtype, accounting for ~75% of all cases [24,25]. Although luminal breast cancer patients, characterized by the presence of estrogen and progesterone receptors (ER and PR), have a better prognosis, recurrence and resistance to endocrine therapy are major challenges [25]. Here, we show that AKT/RICTOR-mediated epigenomic alterations in DNA and histone methylation lead to the rewiring of *UGCG* transcriptional competence in luminal breast cancer (MCF-7 and BT-474) cells. We identify ZFX as the RICTOR-responsive transcription factor that regulates the *UGCG* expression. We further demonstrate that elevated expression of ZFX/UGCG alters ganglioside levels and activates epidermal growth factor receptor (EGFR)-mediated PI3K/AKT/mTOR/MAPK signaling, leading to tumor progression. Finally, Indian patient tumor tissues with high RICTOR expression also manifest high UGCG and ZFX expression, emphasizing the role of mTORC2-regulated ganglioside metabolism in tumor development.

## Results

### Luminal tumors have a deregulated sphingolipid and ganglioside profile and high RICTOR expression

Ceramide-glucosylceramide rheostat plays a crucial role in tumor progression, drug resistance, and chemotherapeutic response, where synthesis of glucosylceramides from ceramides is catalyzed by UGCG, and glucosylceramidase (GBA) hydrolyzes glucosylceramides to ceramides (Fig 1A). To detect any imbalance in the ceramide-glucosylceramide rheostat in luminal breast cancer patients, we quantified the levels of ceramides and glucosylceramides in luminal patient tumor tissues from an Indian cohort and compared them with adjacent normal tissues using liquid chromatography mass spectrometry (LC–MS/MS) in multiple reaction monitoring (MRM) mode (S1 Table) [26,27]. Tumor tissues showed a 2- to 3-fold increase in all ceramide species compared to their adjacent normal tissues (S1A Fig). Similarly, we observed a 2- to 5-fold increase in all glucosylceramide species in tumor tissues (S1B Fig). The glucosylceramide to ceramide ratio for different

species ranged from 1.6- to 4-fold higher in the tumor tissues than in normal tissues, suggesting that the rheostat is shifted towards glucosylceramides in luminal breast cancer patients (Fig 1B).

As glucosylceramides are the starting lipids for gangliosides, we also quantified the key gangliosides (GM3, GM2, GM1, GD3, GD2) of the ganglioside metabolic pathway in luminal tumor tissues and adjacent normal tissues. Interestingly, we observed that luminal tumors have a 2- to 3-fold increase in all GM3 ganglioside species compared to adjacent normal tissues (Fig 1C and 1D). Among disialogangliosides, we observed a 2- to 5-fold increase in all species of GD3 ganglio-sides (except C20:0) (Fig 1C and 1E), whereas only very long chain GD2 gangliosides (C24:1) were elevated in tumor tissues compared to adjacent normal tissues (Fig 1C and 1F). We did not observe any significant change in GM2 gan-gliosides (Fig 1G), whereas C18:0 GM1 ganglioside levels were marginally lower in tumor tissues compared to adjacent normal tissues (Fig 1H). Therefore, these results indicate that luminal tumors have higher GM3 and GD3 gangliosides and lower levels of GM1 gangliosides.

To decipher the effect of increased glucosylceramides on tumor progression, we overexpressed UGCG transcript-encoding cDNA in MCF-7 (MCF-7_UGCG$^{OE}$) and BT-474 (BT-474_UGCG$^{OE}$) cells representing the luminal breast cancer subtype (Figs 1I and S1C). As expected, MCF-7_UGCG$^{OE}$ (3- to 13-fold) and BT-474_UGCG$^{OE}$ cells (1.3- to 1.5-fold) showed an increase in all glucosylceramide species (Figs 1J and S1D). Quantification of gangliosides showed that MCF-7_UGCG$^{OE}$ cells have ~15-fold increased levels of GM3, GD3 (C16:0, C18:0), and GM2 gangliosides and decreased levels of GM1 gangliosides (~3-fold) compared to MCF-7 cells without any change in GD2 gangliosides (Fig 1K–1O). In parallel, BT-474_UGCG$^{OE}$ have ~6-fold increase in GD3 and GM2 gangliosides and diminished levels of GM3 (1.2- to 2-fold) and GM1 gangliosides (~3-fold) compared to BT-474 cells (S1E–S1H Fig). As gangliosides play a crucial role in cell proliferation, MCF-7_UGCG$^{OE}$ and BT-474_UGCG$^{OE}$ cells exhibited elevated cell proliferation (~1.5-fold) compared to MCF-7_VECT$^{OE}$ and BT-474_VECT$^{OE}$ cells (Figs 1P and S1I). In contrast, UGCG silencing showed a decrease (~1.8-fold) in cell proliferation in MCF-7 and BT-474 cells (S1J–S1M Fig). Mice xenograft studies recorded a significant increase in tumor growth kinetics of MCF-7_UGCG$^{OE}$ (~2-fold) and BT-474_UGCG$^{OE}$ (~4-fold) tumors compared to MCF-7_VECT$^{OE}$ and BT-474_VECT$^{OE}$ tumors (Figs 1Q and S1N).

We then investigated the effect of UGCG overexpression on the mTORC1/2 signaling pathway to decipher the mech-anism of UGCG-mediated enhanced cell proliferation in MCF-7_UGCG$^{OE}$ and BT-474_UGCG$^{OE}$ cells. We overexpressed UGCG transcript (with skipped exon 7)-encoding cDNA in MCF-7 (MCF-7_UGCG$^{DEAD}$) and BT-474 (BT-474_UGCG$^{DEAD}$) cells so that they overexpress inactive (dead) UGCG, as supported by attenuated levels of glucosylceramides (S1O and S1P Fig) [28]. Immunoblot studies revealed enhanced expression of RICTOR, pAKT$^{S473}$, and pSGK$^{S78}$ in MCF-7_UGCG$^{OE}$ cells compared to MCF-7_UGCG$^{DEAD}$ cells, whereas there was no significant change in RAPTOR and mTORC1 tar-gets p4EBP1 and p70S6K (Fig 1R). Similarly, we observed enhanced expression of RICTOR, pAKT$^{S473}$, and pSGK$^{S78}$ in BT-474_UGCG$^{OE}$ cells compared to BT-474_UGCG$^{DEAD}$ cells without any significant change in RAPTOR, p4EBP1, and p70S6K (S1Q Fig). The activation of the PI3K/AKT/mTOR pathway on UGCG overexpression hinted at a complex feed-forward loop involving sphingolipid metabolism, RTKs, and PI3K/AKT/mTOR signaling.

RICTOR being a key component of mTORC2, earlier studies on the analysis of The Cancer Genome Atlas (TCGA) curated invasive breast carcinoma patient datasets showed a significant correlation of *RICTOR* upregulation, mutation, or amplification with low overall survival [29,30]. To validate any deregulation of RICTOR expression in luminal breast tumors, we evaluated the RICTOR expression by immunoblotting in tumor and adjoining matched normal breast tissues from luminal cancer patients (S1 Table). We observed an abundant increase in RICTOR expression and high expression of pAKT and UGCG in tumor tissues (Fig 1S). We further validated the enhanced RICTOR expression in luminal tumor tissues by immunofluorescence (S1R Fig). The above results show that luminal breast cancer tissues have high levels of glucosylceramides, high RICTOR, high pAKT, and high UGCG expression. Therefore, we hypothesized that RICTOR (mTORC2) might regulate the sphingolipid metabolism, and increased levels of glucosylceramides may be instrumental for enhanced cancer cell proliferation and tumor progression in luminal breast cancer patients (Fig 1T).

## RICTOR silencing inhibits cell proliferation via UGCG regulation

As luminal tumor tissues show high RICTOR expression, we determined the effect of RICTOR silencing in MCF-7 and BT-474 cells. We generated *RICTOR*-silenced MCF-7_RICTOR$^{SH}$ and MCF-7_SCRAM$^{SH}$ cells with scrambled shRNA as control (Fig 2A), and validated the effect of RICTOR silencing in MCF-7 cells by quantifying the expression of downstream effectors of the mTORC2 pathway. As expected, there was a decrease in the expression of pAKT$^{S473}$ and pSGK$^{S78}$ upon RICTOR silencing, and there was no change in the expression of mTORC1-specific p4EBP1$^{T37}$ and p70S6K (Fig 2B). Cellular assays showed a ~1.25-fold decrease in the proliferation of MCF-7_RICTOR$^{SH}$ cells after 72 h (Fig 2C). Similarly, we observed a decrease in the expression of pAKT$^{S473}$ and pSGK$^{S78}$ upon RICTOR silencing in BT-474-RICTOR$^{SH}$ cells (S2A and S2B Fig) along with a 1.35-fold decrease in cell proliferation after 72 h (S2C Fig). MCF-7_RICTOR$^{SH}$ tumors showed a >3-fold decrease in the kinetics of growth compared to MCF-7_SCRAM$^{SH}$ tumors (Fig 2D). Similarly, BT-474_RICTOR$^{SH}$ tumors showed a ~4-fold lower tumor volume than BT-474_SCRAM$^{SH}$ tumors (S2D Fig).

To decipher the effect of RICTOR silencing on sphingolipid metabolism, we performed quantitative estimation of ceramides and glucosylceramides in MCF-7 and MCF-7_RICTOR$^{SH}$ cells by LC–MS/MS. All glucosylceramide species showed a significant decrease in MCF-7_RICTOR$^{SH}$ cells with a concurrent increase in ceramides compared to MCF-7 cells (Fig 2E). There is a 2- to 9-fold increase in ceramides and a 1.3- to 1.7-fold decrease in glucosylceramides in MCF-7_RICTOR$^{SH}$ cells over MCF-7 cells (Fig 2F). As UGCG is responsible for the synthesis of glucosylceramides from ceramides, we observed a 2-fold decrease in *UGCG* expression by qRT-PCR in MCF-7_RICTOR$^{SH}$ cells compared to MCF-7_SCRAM$^{SH}$ cells (Fig 2G) that was also validated by immunoblotting (Fig 2H). Similarly, we observed a ~1.3-fold decrease in glucosylceramides in BT-474_RICTOR$^{SH}$ cells compared to BT-474_SCRAM$^{SH}$ cells (S2E Fig) along with a ~1.6-fold decrease in *UGCG* expression by qRT-PCR (S2F Fig) and immunoblotting (S2G Fig). In contrast, we did not observe any change in the expression of GBA1, which hydrolyzes glucosylceramides to ceramides (Figs 2H and S2G). MCF-7_RICTOR$^{SH}$ cells showed a decrease in GM3 (2- to 3-fold), GD3 (~55-fold), GM2 (~2-fold), and GM1 (~2.6-fold) gangliosides with no change in GD2 gangliosides compared to MCF-7 cells, suggesting that low influx of glucosylceramides (due to decreased UGCG expression) lowers gangliosides (Fig 2I–2M). Similarly, BT-474_RICTOR$^{SH}$ cells showed ~1.5-fold decrease in GM3 and GD3 gangliosides with no change in GM2 gangliosides compared to BT-474 cells (S2H–S2J Fig).

To confirm that RICTOR-mediated regulation of UGCG expression is responsible for enhanced cell proliferation, we overexpressed UGCG in MCF-7_RICTOR$^{SH}$ cells and observed a ~2-fold increase in glucosylceramides and a ~1.5-fold increase in cell proliferation on UGCG overexpression (Fig 2N–2P). As the proliferation kinetics of MCF-7_RICTOR$^{SH}$ cells on UGCG overexpression do not exactly match with wild-type MCF-7 cells, RICTOR-mediated alteration in the sphingolipid metabolic pathway emerges as one of the key factors regulating cell proliferation, and other factors might also contribute to the RICTOR-mediated effect on cell proliferation. Similarly, overexpression of UGCG in BT-474_RICTOR$^{SH}$ cells caused a ~2-fold increase in cell proliferation (S2K–S2M Fig). To confirm if RICTOR-mediated UGCG regulation exists in other cell types, we silenced RICTOR in HCT-116 (human colon cancer cells) and HEK-293 (human embryonic kidney) cells and observed a decrease in glucosylceramides (S2N–S2Q Fig).

Therefore, these results suggest that the decrease in glucosylceramides upon RICTOR silencing is due to the deregulation of UGCG at the transcriptional level, which further modulates cell proliferation and tumor progression. We now asked how RICTOR regulates the *UGCG* expression and how UGCG-mediated glucosylceramide synthesis enhances tumor progression (Fig 2Q).

## RICTOR regulates UGCG expression via transcription factor Zinc Finger X-linked (ZFX)

We argued that if RICTOR silencing in MCF-7 cells downregulates *UGCG* expression, this regulation may be mediated through transcription factors downregulated upon RICTOR silencing. Therefore, we performed differential gene expression analysis on mouse *RICTOR* (−) microarray datasets from NCBI GEO and identified the transcription factors that

RICTOR silencing inhibits cell proliferation via UGCG regulation.

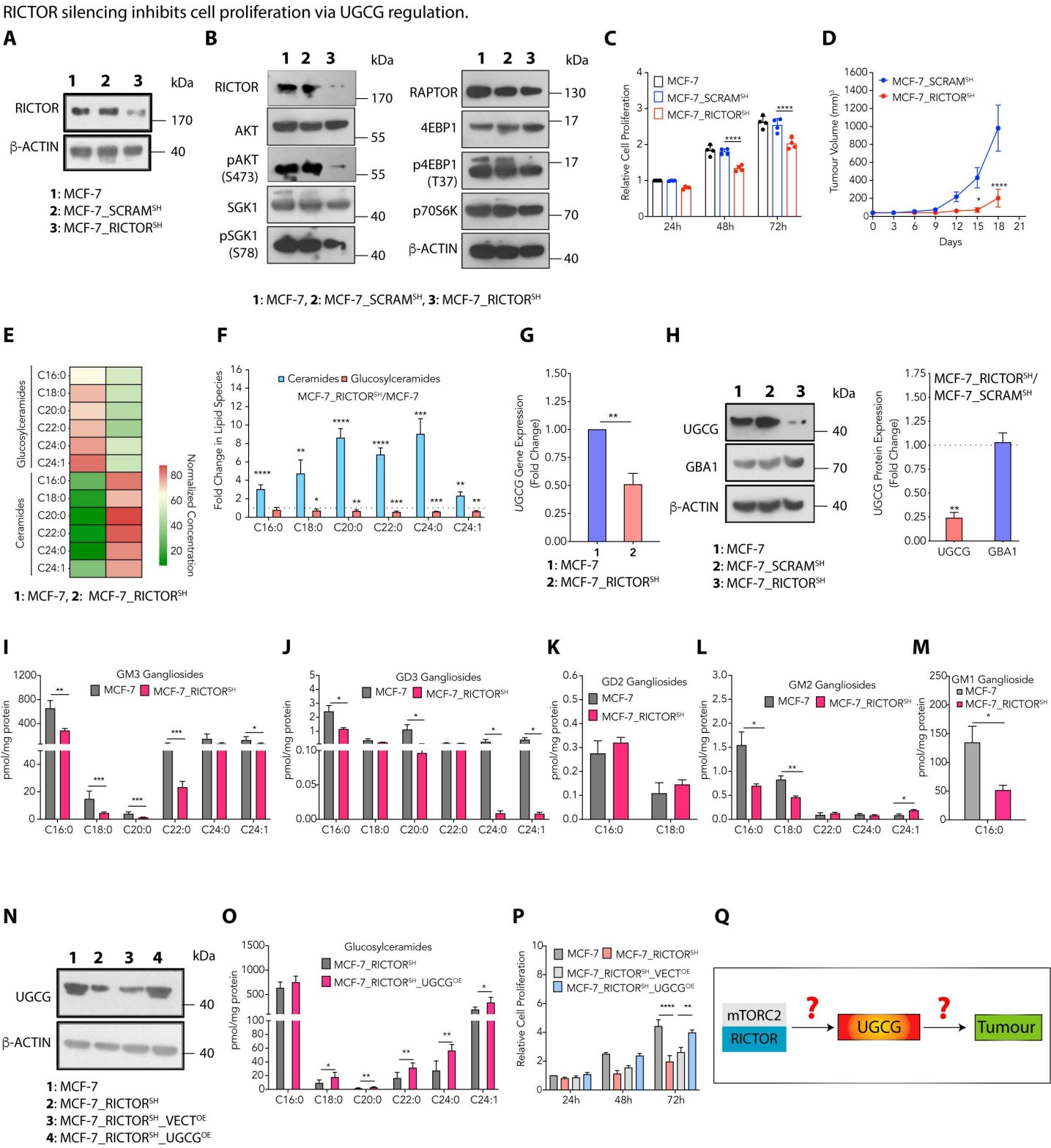

PLOS Biology

**Fig 2. RICTOR silencing inhibits cell proliferation via UCGC regulation.** (**A**) Immunoblots confirm knockdown of RICTOR expression in MCF-7_RICTOR[SH] cells. (**B**) Immunoblots show changes in expression of RICTOR, RAPTOR, and their downstream effectors in MCF-7_RICTOR[SH] cells compared to MCF-7_SCRAM[SH] cells. (**C**) Cell proliferation studies show a decrease in the proliferation of MCF-7_RICTOR[SH] cells (mean±SEM, $n=4$) compared to MCF-7_SCRAM[SH] cells. (**D**) Tumor growth kinetics show significantly slower growth of MCF-7_RICTOR[SH] (mean±SEM, $n=5$) tumors compared to the MCF-7_SCRAM[SH] tumors. (**E**) Heat map representing normalized absolute quantitation of ceramides and glucosylceramides in MCF-7_RICTOR[SH] and MCF-7 cells. (**F**) Fold change (mean±SEM, $n=5$) in different sphingolipid species reveals an increase in ceramides and a decrease in glucosylceramides in MCF-7_RICTOR[SH] cells compared to MCF-7 cells. (**G**, **H**) qRT-PCR (mean±SEM, $n=4$) (**G**) and immunoblots and their quantification (mean±SEM, $n=3$) (**H**) demonstrate downregulation of UCGC without any change in GBA1 expression in MCF-7_RICTOR[SH] cells compared to MCF-7_SCRAM[SH] cells. (**I–M**) Absolute quantification (mean±SEM, $n=3$) of GM3 (I), GD3 (J), GD2 (K), GM2 (L), and GM1 (M) ganglioside species shows a decrease in GM3, GD3, GM2, and GM1 gangliosides in MCF-7_RICTOR[SH] cells compared to MCF-7 cells. (**N**) Immunoblot confirming UCGC overexpression in MCF-7_RICTOR[SH] cells. (**O**) The absolute quantification of glucosylceramides (mean±SEM, $n=4$) in MCF-7_RICTOR[SH]_UCGC[OE] cells compared to MCF-7_RICTOR[SH] cells confirms an increase in glucosylceramides. (**P**) Cell proliferation assay demonstrates an increase in cell proliferation (mean±SEM, $n=4$) of MCF-7_RICTOR[SH] cells on UCGC overexpression. (**Q**) A schematic diagram showing the role of putative factors modulating the RICTOR/pAKT-mediated UCGC expression that can lead to altered glucosylceramides, thereby controlling tumor progression. Data among groups were analyzed using an unpaired Student $t$ test or One-way ANOVA among multiple groups or by Two-way ANOVA in time-dependent studies. $p$-value: *$p<0.05$, **$p<0.01$, ***$p<0.0005$, ****$p<0.0001$. Numerical data can be found in S2 Data.

are downregulated on RICTOR knockdown (Fig 3A) [31]. Using bioinformatic analysis, we also identified experimentally validated transcription factors that bind to the UCGC promoter in MCF-7 cells (Fig 3A) [32]. We found three common transcription factors, Zinc Finger X-linked (*ZFX*), ETS transcription factor (E-74), Like Factor 1 (*ELF1*), and CCCTC-binding Factor (*CTCF*) from this analysis that may bind to the UCGC promoter, and are transcriptionally downregulated on RICTOR silencing. MCF-7_RICTOR[SH] and BT-474_RICTOR[SH] cells showed reduced expression of all these transcription factors by qRT-PCR (Figs 3B and S3A). We further validated the binding of ZFX on *UCGC* promoter by ChIP-qPCR and observed a~1.5-fold decrease in recruitment of ZFX on *UCGC* promoter in MCF-7_RICTOR[SH] and BT-474_RICTOR[SH] cells compared to MCF-7 and BT-474 cells (Figs 3C, S3B, and S3C). The binding of ZFX on UCGC promoter was also validated by Electrophoretic Mobility Shift Assay (EMSA), that showed an increased binding of ZFX on UCGC promoter in MCF-7_ZFX[OE] and BT-474_ZFX[OE] cells compared to MCF-7_VECT[OE] and BT-474_VECT[OE]. (Figs 3D and S3D). Further, ZFX-DNA complexes were validated by shift ablation and competition assay using specific and unrelated oligos in excess. We also compared the endogenous binding of ZFX to UCGC promoter in MCF-7_RICTOR[SH] and BT-474_RICTOR[SH] cells, which showed reduced binding in comparison to MCF-7_SCRAM[SH] and BT-474_SCRAM[SH] cells (Figs 3E and S3E). In line with this, MCF-7_RICTOR[SH] cells showed a decrease in ZFX protein expression (~4-fold) compared to MCF-7 cells (Fig 3F and 3G). Similarly, BT-474_RICTOR[SH] cells showed ~2-fold decrease in ZFX expression compared to BT-474 cells (S3F and S3G Fig).

ZFX is a highly conserved Zinc finger protein and oncogenic transcription factor residing on the X chromosome and is overexpressed in many cancers [33–35]. To functionally validate the role of ZFX in UCGC regulation, we used ZFX-overexpressed MCF-7 cells (MCF-7_ZFX[OE]) and only vector-overexpressed (MCF-7_VECT[OE]) cells as controls. We also performed siRNA-mediated ZFX-silencing in MCF-7 cells (MCF-7_ZFX[SL]) and used scrambled siRNA-transfected (MCF-7_SCRAM[SL]) cells as controls. Immunoblots confirmed the overexpression of ZFX in MCF-7_ZFX[OE] cells and downregulation of ZFX in MCF-7_ZFX[SL] cells (Fig 3H and 3I). As expected, MCF-7_ZFX[OE] cells showed a~2-fold increase, and MCF-7_ZFX[SL] cells showed a~4-fold decrease in UCGC protein expression (Fig 3J and 3K). Similarly, we observed that siRNA-mediated ZFX silencing in BT-474 cells downregulated UCGC expression (S3H Fig), and overexpression of ZFX enhanced the UCGC expression (S3I Fig).

Quantitative estimation of ceramides and glucosylceramides showed a 1.4- to 1.8-fold decrease in ceramides and a 2- to 7-fold increase in glucosylceramides in MCF-7_ZFX[OE] cells compared to MCF-7_VECT[OE] cells (Fig 3L and 3M). In contrast, we observed a 1.8- to 3.8-fold increase in ceramides and a 1.5- to 3-fold decrease in glucosylceramides on ZFX silencing (Fig 3L and 3M). Similarly, we observed an increase in the glucosylceramide to ceramide ratio in BT-474_ZFX[OE] cells compared to BT-474_VECT[OE] cells (S3J Fig). Quantification of gangliosides showed that

RICTOR regulates UGCG expression via transcription factor Zinc Finger-X linked (ZFX).

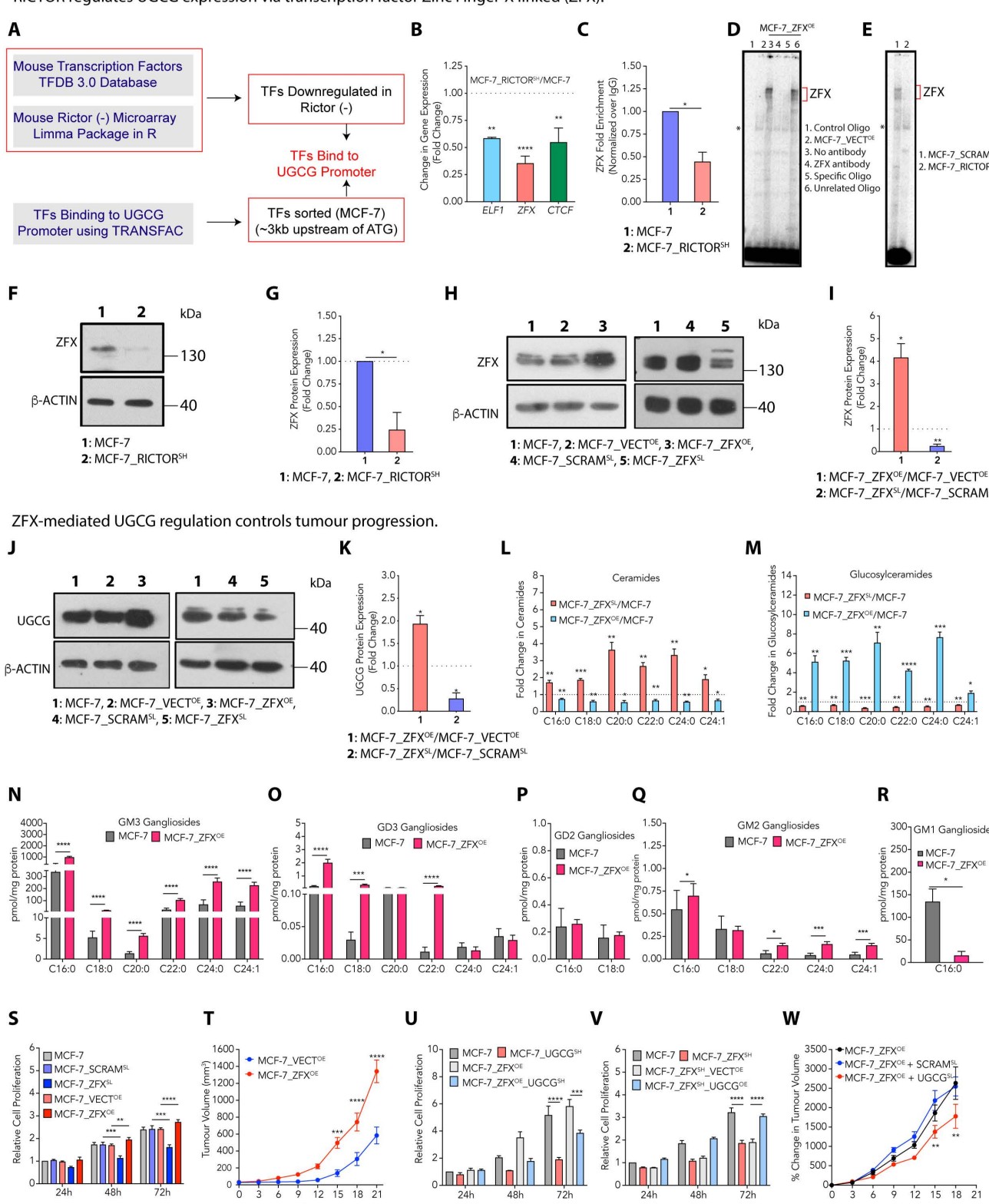

ZFX-mediated UGCG regulation controls tumour progression.

**Fig 3. RICTOR regulates UGCG expression via transcription factor Zinc Finger X-linked (ZFX). (A)** A schematic diagram showing the workflow to identify RICTOR-regulated transcription factors that bind to the UGCG promoter. **(B)** Results from qRT-PCR (mean ± SEM, *n* = 3) confirm reduced

expression of RICTOR-regulated *ELF1*, *ZFX*, and *CTCF* transcription factors in MCF-7_RICTOR[SH] cells. **(C)** ChIP-qPCR (mean ± SEM, *n* = 3) results show reduced binding of ZFX to UGCG promoter in MCF-7_RICTOR[SH] cells. **(D)** EMSA shows the binding of ZFX to UGCG promoter (lanes 2 and 3) in MCF-7_ZFX[OE] cells, shift-ablation assay in MCF-7_ZFX[OE] cells (lane 4), competition assay with specific (lane 5) and unrelated oligo as a control (lane 6). "*" denotes nonspecific complexes. **(E)** EMSA comparing endogenous ZFX-DNA binding activity in MCF-7_SCRAM[SH] and MCF-7_RICTOR[SH] cells. **(F, G)** Immunoblots (F) and their quantification (mean ± SEM, *n* = 3) (G) confirm downregulation of ZFX in MCF-7_RICTOR[SH] cells. **(H, I)** Immunoblots (H) and their quantification (mean ± SEM, *n* = 3) (I) confirm overexpression and silencing of ZFX in MCF-7_ZFX[OE] and MCF-7_ZFX[SL] cells. **(J, K)** Immunoblots (J) and their quantification (mean ± SEM, *n* = 3) (K) show upregulation and downregulation of UGCG upon overexpression and silencing of ZFX in MCF-7_ZFX[OE] and MCF-7_ZFX[SL] cells. **(L, M)** Fold change (mean ± SEM, *n* = 5) in ceramides (L) and glucosylceramides (M) confirms a decrease in ceramides and an increase in glucosylceramides in MCF-7_ZFX[OE] cells. In contrast, MCF-7_ZFX[SL] cells show higher ceramides and reduced glucosylceramides. **(N–R)** Absolute quantification (mean ± SEM, *n* = 3-5) of GM3 (N), GD3 (O), GD2 (P), GM2 (Q), and GM1 (R) gangliosides shows an increase in GM3, GD3, and GM2 gangliosides and attenuated GM1 gangliosides on ZFX overexpression in MCF-7 cells. **(S)** Cell proliferation (mean ± SEM, *n* = 4) demonstrates increased proliferation of MCF-7_ZFX[OE] cells, whereas MCF-7_ZFX[SL] cells show reduced cell proliferation. **(T)** Tumor growth kinetics recorded a significantly higher growth of MCF-7_ZFX[OE] (mean ± SEM, *n* = 4-6) than MCF-7_VECT[OE] tumors. **(U, V)** Cell proliferation demonstrates a decrease in proliferation of MCF-7_ZFX[OE] cells on UGCG silencing (U) (mean ± SEM, *n* = 4), whereas MCF-7_ZFX[SH] cells show enhanced cell proliferation on UGCG overexpression (mean ± SEM, *n* = 3) (V). **(W)** siRNA-mediated silencing of UGCG leads to reduced tumor growth kinetics in MCF-7_ZFX[OE] tumors. Data among two groups were analyzed using an unpaired Student *t* test, among multiple groups using One-way ANOVA, and by Two-way ANOVA in time-dependent studies. *p*-value: \**p* < 0.05, \*\**p* < 0.01, \*\*\**p* < 0.0005, \*\*\*\**p* < 0.0001. Numerical data can be found in S3 Data.

MCF-7_ZFX[OE] cells have an increase of GM3 (~25-fold), GD3 (~20-fold), and GM2 (~4-fold) gangliosides, whereas GM1 gangliosides were downregulated (~9-fold) compared to MCF-7 cells and GD2 gangliosides were not altered significantly (Fig 3N–3R). Interestingly, BT-474_ZFX[OE] recorded a decrease (~40-fold) in GM3 gangliosides along with a steep increase in GD3 gangliosides (>200-fold) (S3K and S3L Fig), thereby suggesting the rheostat shifting towards GD3 gangliosides. There was a ~2.3-fold increase in GM2 gangliosides with a significant decrease in GM1 gangliosides (S3M and S3N Fig).

To elucidate the effect of ZFX-mediated UGCG regulation on cell proliferation and tumor progression, we compared the cell proliferation rates of MCF-7_ZFX[OE] and MCF-7_ZFX[SL] cells. MCF-7_ZFX[OE] exhibited a significant increase in cell proliferation compared to MCF-7_VECT[OE] cells, and ZFX silencing showed a >1.4-fold decrease in proliferation compared to MCF-7_SCRAM[SL] cells after 72 h (Fig 3S). Similarly, we observed enhanced cell proliferation on ZFX overexpression and a significant decrease in cell proliferation on ZFX silencing in BT-474 cells (S3O and S3P Fig). Animal studies recorded a significantly higher growth rate of MCF-7_ZFX[OE] (~2.3-fold) and BT-474_ZFX[OE] (~1.8-fold) tumors than MCF-7_VECT[OE] and BT-474_VECT[OE] tumors (Figs 3T and S3Q).

To validate that ZFX overexpression enhances cell proliferation via regulating UGCG expression, we determined the cell proliferation in UGCG-silenced MCF-7_ZFX[OE] and BT-474_ZFX[OE] cells and compared it with UGCG-silenced MCF-7 and BT-474 cells. Though MCF-7_ZFX[OE]_UGCG[SL] and BT-474_ZFX[OE]_UGCG[SL] cells showed decreased proliferation, this inhibition in proliferation was not as pronounced as in UGCG-silenced MCF-7 and BT-474 cells, suggesting that UGCG is one of the targets of ZFX (Figs 3U and S3R). We also overexpressed UGCG in MCF-7_ZFX[SH] and BT-474_ZFX[SH] cells, and observed enhanced cell proliferation compared to ZFX-silenced cells (Figs 3V and S3S). To validate that increased proliferation on UGCG overexpression in ZFX-silenced cells is due to rescue in gangliosides, we quantified the gangliosides in MCF-7_ZFX[SH]_UGCG[OE] and BT-474_ZFX[SH]_UGCG[OE] cells and recorded the rescue of GD3 gangliosides compared to MCF-7_ZFX[SH] and BT-474_ZFX[SH] cells (S3T Fig). Animal studies further validated that silencing of UGCG in MCF-7_ZFX[OE] tumors led to a decrease (1.5-fold) in the tumor growth of MCF-7_ZFX[OE] tumors (Fig 3W). To see if ZFX-mediated UGCG regulation is a general phenomenon, we silenced ZFX in HCT-116, HEK-293, HepG2, and MDA-MB-453 cells and observed significant downregulation of UGCG expression (S3U Fig). These results affirm that ZFX, regulated by RICTOR, is one of the transcription factors that modulate UGCG expression in luminal cancer cells, priming the increase in glucosylceramides and enhancing tumor progression. Therefore, the next step was to delineate how RICTOR controls UGCG expression.

## DNMT1 regulates ZFX-mediated UGCG transcription

pAKT-mediated regulation of DNA methyltransferases (DNMTs), histone demethylases (HDMs), and histone methyltransferases (HMTs) can lead to epigenomic alterations and influence gene transcription [36]. Tumor tissues carry higher DNA hypomethylation compared to normal tissues, and this DNA hypomethylation contributes to cancer progression through activation of gene expression, transactivation of transposable elements, and chromosomal instability, thereby causing enhanced proliferation and evasion of apoptosis [37]. Earlier studies have shown that AKT-mediated phosphorylation of DNMT1 disrupts the DNMT1/PCNA/UHRF1 interactions, leading to gene-specific hypomethylation [38]. As the UGCG promoter has multiple CpG islands, and ZFX is known to bind at CpG islands in gene promoters [39], we hypothesized that pAKT-mediated phosphorylation of DNMT1 promotes demethylation of CpG islands and enhances the binding of ZFX to the *UGCG* promoter [36] (Fig 4A). Therefore, we performed an immunoprecipitation (IP)-western assay for DNMT1 and checked for its altered phosphorylation levels using pan-phospho-serine-antibody in IP fractions. Immunoblots clearly showed that RICTOR silencing inhibited the phosphorylation of DNMT1 in MCF-7 and BT-474 cells, and overexpression of ZFX enhanced the phosphorylation of DNMT1 (Figs 4B and S4A). To validate that pAKT-mediated phosphorylation of DNMT1 is necessary and essential for UGCG expression, we tested the impact of pAKT inhibitor (MK2206) on UGCG expression in MCF-7 and BT-474 cells and observed a dose-dependent decrease in UGCG expression at the translational (Figs 4C and S4B) and transcriptional level (Figs 4D and S4C). IP assay for DNMT1 in MCF-7_ZFX$^{OE}$ and BT-474_ZFX$^{OE}$ cells, followed by immunoblotting with pan-phospho-serine-antibody, confirmed a decrease in phosphorylation of DNMT1 in the presence of AKT inhibitor, MK2206 (Figs 4E and S4D). These results indicate that elevated pAKT-mediated phosphorylation of DNMT1, at least in part, regulates the methylation status of the *UGCG* promoter, thereby allowing ZFX-mediated UGCG expression.

To support the DNMT-mediated UGCG regulation, we inhibited DNMTs in MCF-7_RICTOR$^{SH}$ and BT-474-RICTOR$^{SH}$ cells by decitabine (DAC) treatment (2, 5, 10 µM) and found a 2- to 15-fold increase in *UGCG* expression by qRT-PCR (Figs 4F and S4E). Immunoblot studies confirmed an increase in UGCG expression in MCF-7_RICTOR$^{SH}$ and BT-474_RICTOR$^{SH}$ cells on DAC treatment (Figs 4G and S4F). Using ChIP-qPCR, we witnessed a 2-fold increase in ZFX enrichment and binding to the *UGCG* promoter in DAC-treated MCF-7_RICTOR$^{SH}$ and BT-474_RICTOR$^{SH}$ cells (Figs 4H and S4G). DAC-treated MCF-7_RICTOR$^{SH}$ and BT-474_RICTOR$^{SH}$ cells also showed ~1.3-fold increased cell proliferation (S4H and S4I Fig). Therefore, these studies demonstrate that pAKT-mediated phosphorylation of DNMT1 in MCF-7 and BT-474 cells or DAC-mediated downregulation of DNMTs in MCF-7_RICTOR$^{SH}$ and BT-474_RICTOR$^{SH}$ cells enhances the recruitment of ZFX to *UGCG* promoter and elevates UGCG expression.

## RICTOR regulates UGCG expression via histone demethylase KDM5A

Promoter-associated trimethylation of histone H3 (H3K4Me3) is one of the key targets of PI3K/AKT and acts as a mode of regulating transcriptional competence [40]. H3K4Me3 status is regulated by histone demethylase, KDM5A [41]. AKT-mediated phosphorylation of KDM5A is instrumental in its nuclear exit, leading to elevated H3K4Me3 levels, thereby augmenting gene transcription (Fig 4I). Therefore, we performed an IP-Western assay for KDM5A and checked its altered phosphorylation levels using a pan-phospho-serine antibody in IP fractions. Immunoblots suggested that RICTOR silencing inhibited the phosphorylation of KDM5A in MCF-7 and BT-474 cells (Figs 4J and S4J). As phosphorylation of KDM5A will cause its nuclear exit, we quantified the KDM5A in cytosolic and nuclear extracts of MCF-7 and MCF-7_RICTOR$^{SH}$ cells and observed that RICTOR silencing led to a >4-fold increase in retention of KDM5A in the nucleus (Fig 4K and 4L). Similarly, RICTOR silencing enhanced the retention of KDM5A in the nucleus in BT-474 cells (S4K Fig).

To validate that pAKT-mediated phosphorylation of KDM5A is necessary and essential for UGCG expression, we performed an IP assay for KDM5A followed by immunoblotting with pan-phospho-serine-antibody in MCF-7_ZFX$^{OE}$ and BT-474_ZFX$^{OE}$ cells. The results confirmed the decrease in phosphorylation of KDM5A on treatment with pAKT inhibitor

DNMT1 regulates ZFX-mediated UGCG transcription.

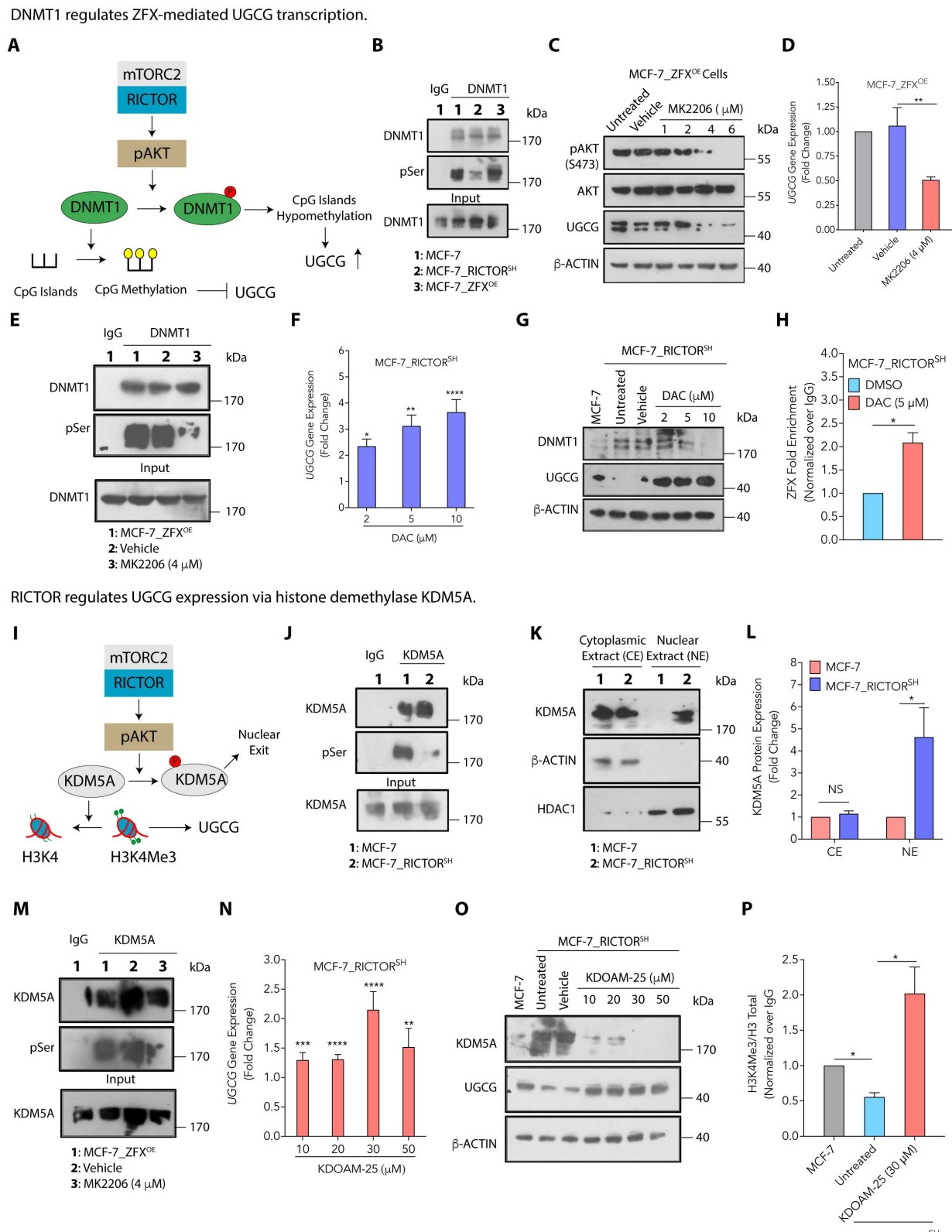

RICTOR regulates UGCG expression via histone demethylase KDM5A.

**Fig 4. AKT regulates UGCG expression via epigenomic alterations.** (**A**) A schematic diagram showing AKT-mediated phosphorylation of DNMT1 leading to hypomethylation of CpG islands that further enhances UGCG expression. (**B**) Immunoblot showing alteration in phosphorylation of DNMT1

by pan-phospho-ser antibody in MCF-7_RICTOR[SH] and MCF-7_ZFX[OE] cells compared to MCF-7 cells. (C) Immunoblots showing a dose-dependent decrease in pAKT and UGCG expression in MCF-7_ZFX[OE] cells on treatment with AKT inhibitor MK2206. (D) Results from qRT-PCR (mean±SEM, $n=4$) show a decrease in *UGCG* expression in MCF-7_ZFX[OE] cells on AKT inhibition by MK2206. (E) Immunoblot reveals a decrease in phosphorylation of DNMT1 on treatment of MCF-7_ZFX[OE] cells with AKT inhibitor MK2206. (F, G) Results from qRT-PCR (mean±SEM, $n=3$) (F) and immunoblot (G) show increased *UGCG* expression in MCF-7_RICTOR[SH] cells on DNMT inhibition by DAC. (H) ChIP-qPCR results (mean±SEM, $n=3$) confirm enhanced binding of ZFX to UGCG promoter in MCF-7_RICTOR[SH] cells on DAC (5 μM) treatment. (I) A schematic representation of pAKT-mediated regulation of histone demethylase KDM5A that regulates UGCG transcription via histone methylation. (J) Immunoblot showing the change in phosphorylation of KDM5A using the pan-phospho-Ser antibody in MCF-7_RICTOR[SH] cells compared to MCF-7 cells. (K, L) Immunoblot (K) and its quantification (mean±SEM, $n=3$) (L) show alterations in KDM5A expression in nuclear and cytoplasmic extracts in MCF-7_RICTOR[SH] cells compared to MCF-7 cells. (M) Immunoblot reveals a decrease in phosphorylation of KDM5A on treatment of MCF-7_ZFX[OE] cells with AKT inhibitor MK2206. (N, O) Results from qRT-PCR (mean±SEM, $n=3$) (N) and immunoblot (O) show increased *UGCG* expression in MCF-7_RICTOR[SH] cells on KDM5A inhibition. (P) ChIP-qPCR results (mean±SEM, $n=3$) show a reduced H3K4Me3 mark on UGCG promoter in MCF-7_RICTOR[SH] cells that increases on treatment with KDOAM-25 inhibitor (30 μM). Data among two groups were analyzed using an unpaired Student $t$ test and among multiple groups using One-way ANOVA. $p$-value: *$p<0.05$, **$p<0.01$, ***$p<0.0005$, ****$p<0.0001$. Numerical data can be found in S4 Data.

MK2206 (Figs 4M and S4L). Inhibition of KDM5A using a chemical inhibitor, KDOAM-25, demonstrated an increase in *UGCG* expression by ~1.2- to 2.0-fold in MCF-7_RICTOR[SH] and by ~2.5- to 17-fold in BT-474-7_RICTOR[SH] cells (Figs 4N and S4M). This was validated by immunoblot studies showing an increase in UGCG expression with a concurrent decrease in KDM5A expression in MCF-7_RICTOR[SH] and BT-474-7_RICTOR[SH] cells (Figs 4O and S4N). ChIP-PCR results confirmed that MCF-7_RICTOR[SH] and BT-474-7_RICTOR[SH] cells possess a reduced H3K4Me3 mark compared to MCF-7 and BT-474 cells, and it gets elevated by 2- to 6-fold upon KDM5A inhibition (30 μM) (Figs 4P and S4O). As KDM5A inhibition leads to an increase in UGCG expression, cellular assays confirmed increased cell proliferation in MCF-7_RICTOR[SH] and BT-474_RICTOR[SH] cells on treatment with KDOAM-25 (S4P and S4Q Fig). Therefore, these results confirm that mTORC2-AKT-mediated phosphorylation of KDM5A causes its exit from the nucleus and does not allow demethylation of H3K4Me3, thereby promoting activation of *UGCG* transcription leading to tumor progression. Therefore, AKT/RICTOR-mediated phosphorylation of DNMT1 and/or KDM5A regulates gene expression, including that of UGCG, leading to an increase in glucosylceramides and enhanced tumor progression. To complete the circuit connecting the metabolic-gene regulatory signaling, the next step was to find how UGCG-mediated increase in glucosylceramides enhances tumor progression.

**GD3-mediated EGFR activation drives cell proliferation and tumor progression**

Gangliosides present in GEMs are well-known to regulate RTK signaling, contingent upon the nature and relative quantity of gangliosides, the kind of growth factor receptors, and the cell type [42]. Gangliosides like GD2 and GD3 can activate RTK signaling and enhance cancer cell proliferation [43]. In contrast, gangliosides like GM1 can mitigate RTK signaling by inhibiting the dimerization of growth factor receptors [44]. Glucosylceramides are the starting lipids for ganglioside synthesis, marking UGCG as one of the decisive enzymes that control ganglioside synthesis. As there is an increase in GD3 gangliosides on UGCG/ZFX overexpression and a decrease on RICTOR silencing in MCF-7 and BT-474 cells, there might also be changes in the deregulation of other ganglioside metabolic/catabolic enzymes. Therefore, we evaluated the expression of all ganglioside-metabolic pathway enzymes and observed variable, cell-type-specific alterations in expression of enzymes like neuraminidase (NEU1-4), hexosaminidase (HexA/B), and α/β-galactosidases (GLA/GLB) in MCF-7 and BT-474 cells overexpressing UGCG and ZFX (S5A and S5B Fig). Though expression of some of the catabolic enzymes like NEU3, HEXA/B showed alterations in the UGCG/ZFX overexpressed luminal cell lines in comparison to MCF-7 and BT-474 cells however, since catabolism of gangliosides involves multienzyme complexes including NEU proteins, protective protein Cathepsin A, GM1 cleaving B-galactosidase, GM2 activator protein/Saposin B and others, it is beyond the scope of this study to delve deeper into the dynamics of these deregulations [45]. Expression of other ganglioside metabolic enzymes (ST3GALV, B4GALNT1, B3GALT4, ST8SIA1) was again cell-type specific and exhibited marginal changes in some cases (S5A and S5B Fig).

To study the regulatory effects of ganglioside alterations on RTK signaling, we estimated the expression of phosphorylated epidermal growth factor receptor (EGFR) and its downstream signaling components. Immunoblot analysis showed no alterations in expression of total EGFR in MCF-7_RICTOR$^{SH}$, MCF-7_UGCG$^{OE}$, and MCF-7_ZFX$^{OE}$ cells compared to MCF-7 cells (Fig 5A). However, we observed an increase in phosphorylated EGFR (pEGFR$^{Y1173}$ and pEGFR$^{Y1068}$) in MCF-7_UGCG$^{OE}$ and MCF-7_ZFX$^{OE}$ cells over that of MCF-7 cells (Fig 5A). We also observed activation and upregulation of downstream signaling intermediates, pAKT$^{S473}$ and extracellular signal-regulated protein kinase/p(ERK1/2) in MCF-7_UGCG$^{OE}$ and MCF-7_ZFX$^{OE}$ cells in comparison to MCF-7 cells (Fig 5A). Therefore, these results suggest that over-expression of ZFX or UGCG hyperactivates EGFR-mediated RTK signaling. Similarly, we observed activation of EGFR signaling in BT-474_UGCG$^{OE}$ and BT-474_ZFX$^{OE}$ cells compared to BT-474 cells (S5C Fig).

The above results implied that the UGCG-promoted increase in GD3 gangliosides might be responsible for enhanced pEGFR signaling and cell proliferation in MCF-7 and BT-474 cells. As ST8SIA1 is the key enzyme that synthesizes GD3 gangliosides, we silenced ST8SIA1 in MCF-7 and BT-474 cells and observed a reduced pEGFR$^{Y1173}$ and pEGFR$^{Y1068}$ expression (Figs 5B and S5D). Silencing of ST8SIA1 also decreased the cell proliferation by ~2-fold in MCF-7 and BT-474 cells (Figs 5C and S5E). Further validation was provided on overexpression of ST8SIA1 that enhanced the expression of pEGFR$^{Y1173}$ and pEGFR$^{Y1068}$ (Figs 5D and S5F) and increased the cell proliferation in MCF-7 (~1.7-fold) and BT-474 (~1.9-fold) cells (Figs 5E and S5G). Attenuated GD3 gangliosides in MCF-7_ST8SIA1$^{SH}$ and BT-474_ST8SIA1$^{SH}$ cells and elevated GD3 gangliosides in MCF-7_ST8SIA1$^{OE}$ and BT-474_ST8SIA1$^{OE}$ cells endorsed the role of ST8SIA1-mediated manipulation of GD3 gangliosides in altered cell proliferation (Figs 5F and S5H). We also overexpressed B3GALT4 in MCF-7 cells and observed a decrease in cell proliferation (~2-fold) along with the increase in some species of GM1 gangliosides and a reduction of GD3 gangliosides (Fig 5G–5I). We did not observe reduced cell proliferation on overexpression of B3GALT4 in BT-474 cells (S5I–S5K Fig), which may be due to the exceptionally high levels of GD3 gangliosides in BT-474 cells.

To further validate whether RICTOR-mediated alterations in sphingolipid/ganglioside metabolites contribute to cell proliferation, we treated the MCF-7_RICTOR$^{SH}$ cells with glucosylceramides and different gangliosides cultured in lipid-free media. We observed a >1.5-fold increase in cell proliferation on exogenous treatment with glucosylceramides (Fig 5J). Similarly, the incubation of MCF-7_RICTOR$^{SH}$ cells with GD3 gangliosides caused a >1.3-fold increase in cell proliferation, and treatment with GM1 gangliosides inhibited the cell proliferation (Fig 5J). We did not see any significant alterations in the cell proliferation on exogenous feeding of GM3, GM2, and GD2 gangliosides. Similarly, treatment of BT-474_RICTOR$^{SH}$ cells with GD3 gangliosides caused a ~1.2-fold increase in cell proliferation(S5L Fig). We did not observe any decrease in proliferation on GM1 feeding due to the higher GD3 gangliosides in BT-474 cells. Therefore, these results confirm that GD3-mediated activation of EGFR is responsible, at least partially, for cell proliferation, thus completing the metabolic-signaling-gene regulation circuit.

To further validate the role of GD3 ganglioside-mediated activation of EGFR, we performed siRNA-mediated silencing of GD3 synthase (ST8SIA1) in MCF-7_ZFX$^{OE}$ cells. We observed a significant downregulation in pEGFR$^{Y1173}$ and pEGFR$^{Y1068}$ expression, though the total EGFR was unaltered (Fig 5K). ST8SIA1 silencing using siRNA attenuated the cell proliferation of MCF-7_ZFX$^{OE}$ cells by ~1.25-fold (Fig 5L) and even abrogated the growth kinetics in mice xenografts by ~2-fold (Fig 5M). Similarly, siRNA-mediated ST8SIA1 silencing attenuated the pEGFR$^{Y1173}$ and pEGFR$^{Y1068}$ and cell proliferation of BT-474_ZFX$^{OE}$ cells (S5M and S5N Fig). Therefore, these results suggest that RICTOR regulates cell proliferation by altering the expression of glucosylceramides and gangliosides.

## ZFX expression is strongly associated with UGCG in luminal patients

To find the association between *ZFX* and *UGCG* in luminal breast cancer patients, we analyzed the TCGA-BRCA and METABRIC patient gene expression datasets. The TCGA patient dataset ($N = 1,082$) based on PAM50 classification ($N = 833$) was divided into luminal A ($N = 416$), luminal B ($N = 185$), and non-luminal subtypes ($N = 232$) (Fig 6A)

GD3-mediated EGFR activation drives cell proliferation and tumour progression.

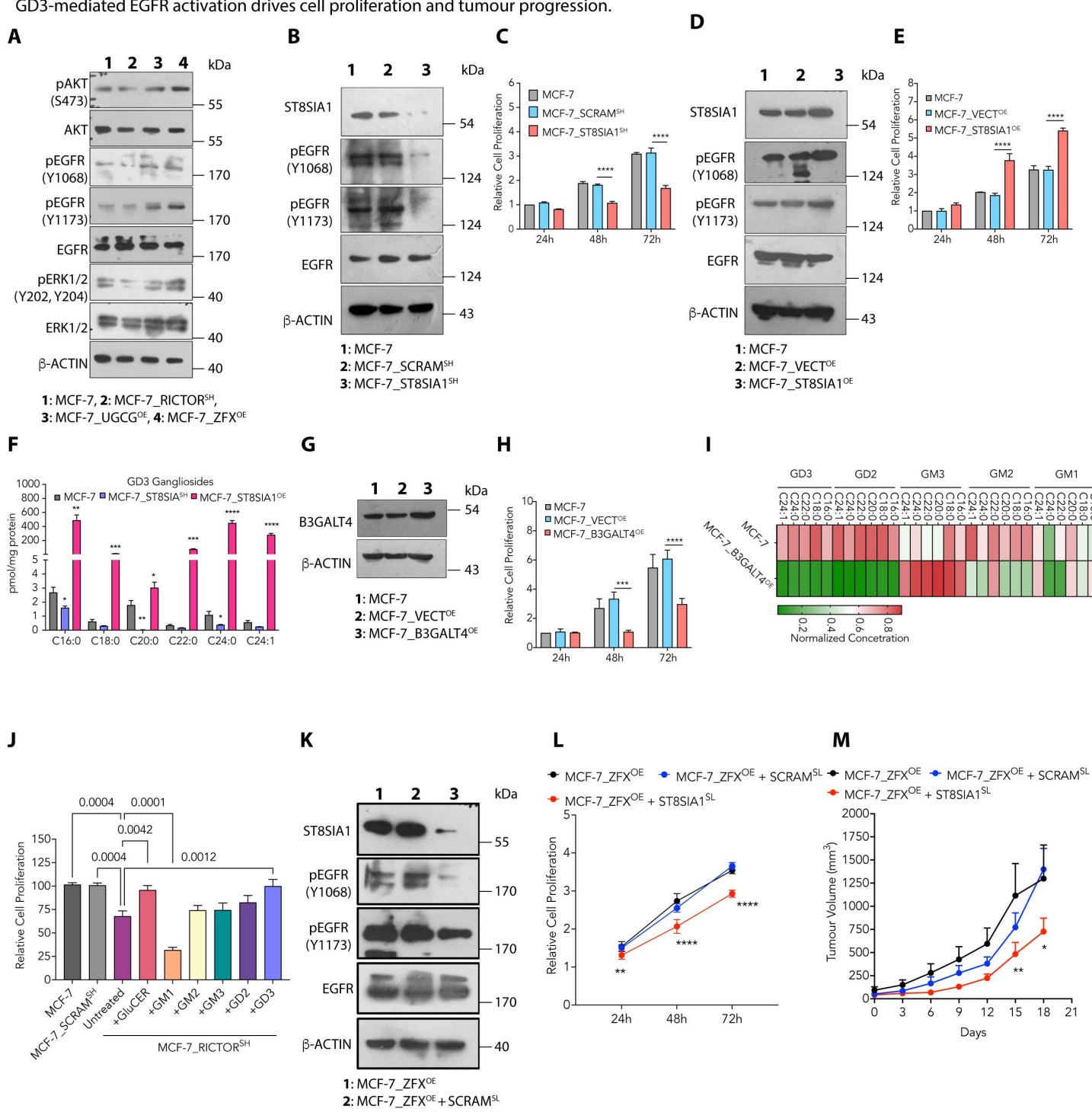

**Fig 5. GD3-mediated EGFR activation drives cell proliferation and tumor progression.** (**A**) Immunoblots reveal an increase in pEGFR[Y1068], pEG-FR[Y1173], pAKT[S473], and pERK1/2[(Y202, Y204)] in MCF-7_UGCG[OE] and MCF-7_ZFX[OE] cells compared to MCF-7 cells. (**B**) Immunoblots show attenuated EGFR activation on shRNA-mediated silencing of GD3 synthase (ST8SIA1) in MCF-7 cells. (**C**) Cell proliferation assay demonstrates a decrease in cell prolif-eration (mean ± SEM, $n = 3$) of MCF-7_ST8SIA1[SH] cells compared to MCF-7 cells. (**D**) Immunoblots show enhanced EGFR activation on overexpression of ST8SIA1 in MCF-7 cells. (**E**) Cell proliferation assay demonstrates increased proliferation (mean ± SEM, $n = 3$) of MCF-7_ST8SIA1[OE] cells compared

to MCF-7 cells. (**F**) Absolute quantification (mean±SEM, $n=3$–4) of GD3 gangliosides validates the silencing and overexpression of ST8SIA1 in MCF-7 cells. (**G**) Immunoblots confirm overexpression of B3GALT4 in MCF-7 cells. (**H**) Cell proliferation assay demonstrates decreased cell proliferation (mean±SEM, $n=4$) of MCF-7_B3GALT4$^{OE}$ cells compared to MCF-7 cells. (**I**) Absolute quantification (mean±SEM, $n=5$) of gangliosides validates the overexpression of B3GALT4. (**J**) Cell proliferation assay (mean±SEM, $n=3$) showing an increase in proliferation of MCF-7_RICTOR$^{SH}$ cells upon supplementing GD3 gangliosides and a decrease in cell proliferation upon feeding with GM1 gangliosides. (**K**) Immunoblots show attenuated EGFR activation on siRNA-mediated silencing of ST8SIA1 in MCF-7_ZFX$^{OE}$ cells. (**L**) Cell proliferation assay demonstrates a decrease in cell proliferation (mean±SEM, $n=4$) of MCF-7_ZFX$^{OE}$ cells on siRNA-mediated inhibition of ST8SIA1. (**M**) Tumor growth kinetics using xenograft studies show a decrease in growth kinetics (mean±SEM, $n=4$–6) of MCF-7_ZFX$^{OE}$ tumors on siRNA-mediated inhibition of ST8SIA1. Data among groups were analyzed using an unpaired Student $t$ test, among multiple groups using One-way ANOVA, and by Two-way ANOVA in time-dependent studies. $p$-value: *$p<0.05$, **$p<0.01$, ***$p<0.001$, ****$p<0.0001$. Numerical data can be found in S5 Data.

[46]. Differential gene expression data analysis showed that luminal A and B tumor tissues have significantly higher expression of *UGCG* (Fig 6B) and *ZFX* (Fig 6C) than basal and HER2$^+$ groups. Correlating *ZFX* and *UGCG* expression to ER and PR status revealed that ER$^+$ (Fig 6D and 6E) and PR$^+$ (Fig 6F and 6G) tumors have significantly higher expression of *UGCG* (Fig 6D and 6F) and *ZFX* (Fig 6E and 6G). Gene expression analysis further revealed that >56% of luminal (luminal A and B) patients have high expression of both *UGCG* and *ZFX* (Fig 6H). Similarly, we divided the METABRIC data sets based on PAM50 classification ($N=1,905$) into luminal A ($N=696$), luminal B ($N=474$), and non-luminal ($N=728$) subtypes (S6A Fig) [47]. Metadata analysis further confirmed that luminal A and luminal B tumor tissues have higher *UGCG* and *ZFX* expression over other subtypes as observed in the TCGA data set (S6B and S6C Fig), and expression of UGCG and ZFX is also high in ER$^+$ and PR$^+$ tumor tissues (S6D–S6G Fig). We identified that ~45% of luminal (luminal A and B) patients with higher *UGCG* expression also exhibit higher *ZFX* expression (S6H Fig). To assess the correlation of *RICTOR*, *UGCG*, and *ZFX* in cancer cells, we analyzed publicly available single-cell RNA sequencing datasets from breast cancer patients (GSE176078). Uniform Manifold Approximation and Projection (UMAP) combined with unsupervised clustering identified nine distinct cellular clusters, including normal and malignant epithelial cells, immune cells, endothelial cells, and various stromal cell types (S6I Fig). Expression analysis revealed that *RICTOR*, *UGCG*, and *ZFX* were highly expressed in cancer cells. While *RICTOR* and *ZFX* showed similar expression in both normal and cancerous epithelial cells, *UGCG* expression was elevated in cancer cells compared to normal epithelial cells. These genes were also expressed in several noncancerous cell types, including endothelial cells, cancer-associated fibroblasts (CAFs), and perivascular-like (PVL) cells, with moderate expression observed in immune cells. (S6J Fig). It is to be noted that scRNA analysis only presents the data at transcript levels from a small set of samples, unlike the large TCGA and METABRIC datasets.

To further validate the association of ZFX and UGCG in breast cancer patients, we quantified the expression of UGCG and ZFX by immunohistochemical (IHC) analysis in tumor samples of all subtypes ($N=90$) (S2 Table). In concurrence with TCGA BRCA and METABRIC datasets, IHC analysis confirmed that ~15% of luminal patients have high ZFX and UGCG expression (Fig 6I and 6J). Finally, we quantified the expression of *UGCG* and *ZFX* from luminal tumors by qRT-PCR, and observed ~2-fold increase in the expression of *ZFX* and *UGCG* in tumor tissues over adjacent matched normal control (Fig 6K). Therefore, these results support a positive correlation between ZFX and UGCG expression in luminal patients.

As UGCG has emerged as a key therapeutic target, we targeted the inhibition of UGCG using the FDA-approved UGCG inhibitor Eliglustat in MCF-7 and BT-474 tumor models upon intraperitoneal delivery. Eliglustat treatment causes a >1.5-fold decrease in tumor volume in MCF-7 tumors (Fig 6L). We validated the effect of eliglustat treatment by quantifying the change in glucosylceramides and gangliosides and observed a significant decrease in glucosylceramides, lactosylceramides, and GD3 gangliosides. (Figs 6M and S6K). Similarly, we observed a >2-fold decrease in BT-474 tumors on eliglustat treatment (Fig 6N). Interestingly, eliglustat treatment did not decrease the levels of glucosylceramides, but we observed a significant decrease in lactosylceramides and GD3 gangliosides (Figs 6O and S6K). As BT-474 is a highly proliferative cell line with high glucosylceramide levels, the decrease in glucosylceramides on Eliglustat treatment is probably compensated through other

ZFX expression is strongly associated with UGCG in luminal patients.

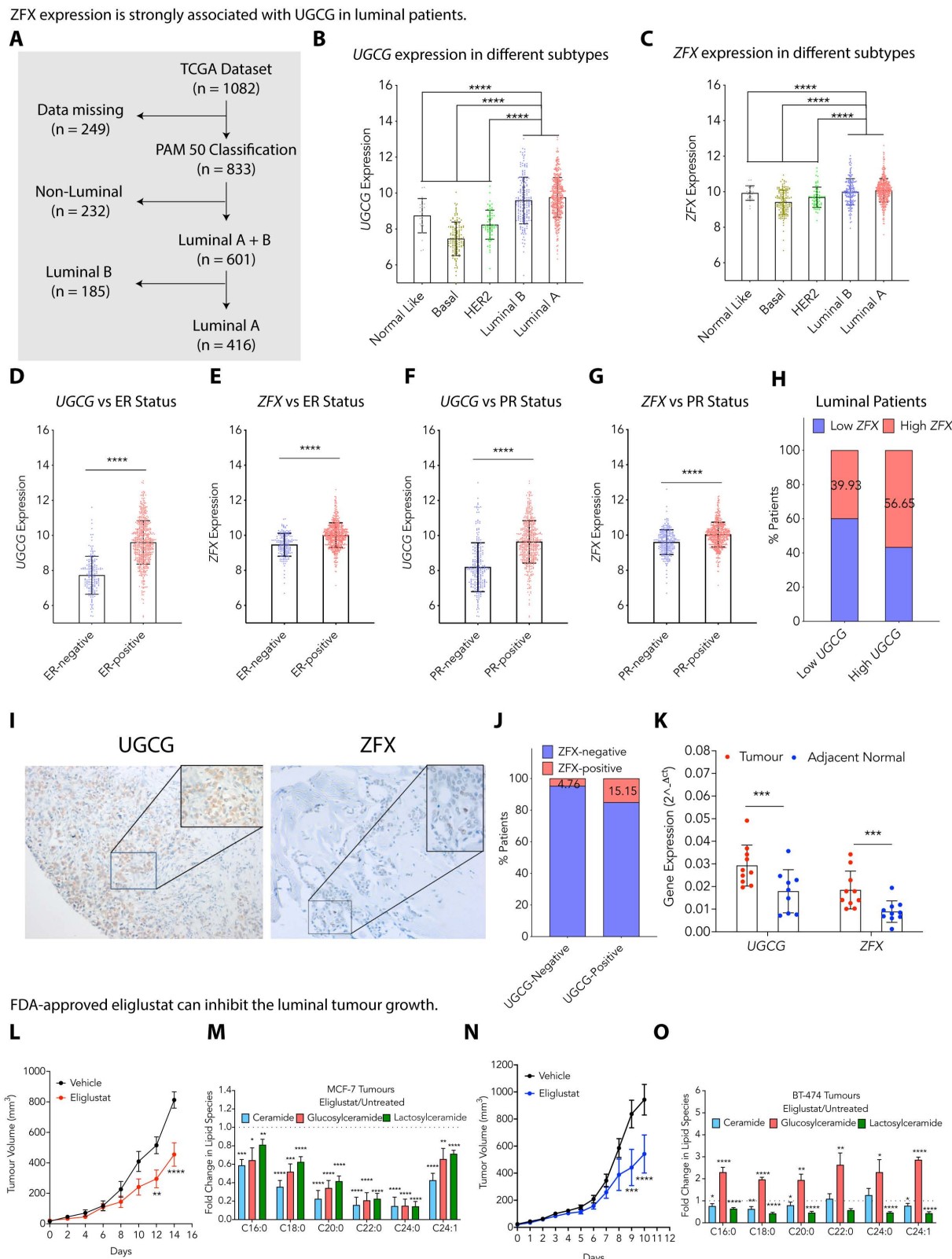

FDA-approved eliglustat can inhibit the luminal tumour growth.

**Fig 6. ZFX expression is strongly associated with UGCG in luminal patients.** (**A**) A schematic diagram showing the PAM50 classification of the TCGA tumor dataset used for analysis. (**B**, **C**) Gene expression of *UGCG* (**B**) and *ZFX* (**C**) in different breast cancer subtypes (PAM50) of the TCGA

dataset confirms high expression of *UGCG* and *ZFX* in luminal subtypes compared to other subtypes. (**D–G**) Change in expression of *UGCG* (D, F) and *ZFX* (E, G) with respect to ER (D, E) and PR (F, G) status in breast tumors of the TCGA dataset confirms high *UGCG* (D, F) and high *ZFX* (E, G) expression in ER+ and PR+ tumors. (**H**) Percentage of tumors having high expression of *UGCG* and *ZFX* among luminal subtype tumors in the TCGA dataset. (**I**) Representative immunohistochemical images show enhanced cytoplasmic UGCG and increased nuclear stain of ZFX in luminal breast tissues. All images are at 100× magnification, and insets are at 400× magnification. (**J**) Percentage of luminal tumors from the Indian cohort (*N*=90) positive for both UGCG and ZFX on immunohistochemical staining. (**K**) qRT-PCR (mean±SEM, *n*=10) validation showing high *UGCG* and high *ZFX* expression from luminal subtype tumors in comparison to adjacent normal tissues in an Indian cohort. (**L**) Change in tumor growth kinetics of MCF-7 tumors on treatment with Eliglustat (Mean±SEM, *n*=5). (**M**) Changes in glucosylceramides and lactosylceramides (Mean±SEM, *n*=4) in eliglustat-treated tumors compared to untreated MCF-7 tumors. (**N**) Change in tumor growth kinetics of BT-474 tumors on treatment with eliglustat (Mean±SEM, *n*=6). (**O**) Changes in glucosylceramides and lactosylceramides (Mean±SEM, *n*=3) in eliglustat-treated tumors compared to untreated BT-474 tumors. Data among two groups were analyzed using an unpaired Student *t* test, among multiple groups using One-way ANOVA, and by Two-way ANOVA in time-dependent studies. *p*-value: \**p*<0.05, \*\**p*<0.01, \*\*\*\**p*<0.0001. Numerical data can be found in S6 Data.

salvage pathways, and therefore does not show the effect. Therefore, these results confirm that UGCG is a potential therapeutic target regulated by RICTOR through epigenetic regulations and can be explored further for cancer treatment.

## Discussion

Treating cancer is like the Herculean duel with the chthonic creature Hydra, whose decapitation magically led to a botanical duplication of the regenerated heads. The myriad of alternative strategies that cancer cells employ to achieve survival advantage over clinical interventions is a similar saga. One of the prime reasons for this is our lack of complete understanding of the metabolic signaling and gene regulatory networks that cancer cells deploy to survive. More important is to understand how these networks are interconnected so that multiple nodes in the circuit/network can be combinatorially targeted to overrule their survival strategies. In this context, herein, we have mapped the first step of the metabolite-cell signaling-gene regulatory circuit connecting ganglioside metabolism with cancer, controlled by the EGFR-mTORC2/RICTOR complex (Fig 7).

The major biological roles of sphingolipids and gangliosides at the cell surface include modulating the lipid phase of cellular membranes, acting as ligands for membrane receptors, kinases, and enzymes, and surface recognition through glycan interactions by glycosphingolipids [48]. Gangliosides, as a part of GEMs, can act as double-edged swords where they can either augment or inhibit the growth factor-mediated cell proliferation through activation/deactivation of RTK signaling cascades [49]. Using elegant precedence of ganglioside-mediated activation of growth factor receptor signaling, our work reveals that increased GD3 gangliosides in response to altered expression of UGCG boost the EGFR phosphorylation status and subsequent downstream growth signaling in luminal cancers. Although RTK inhibitors have made breakthroughs in tumor treatment options, RTK co-activation networks mark a serious limitation in their use [50]. This effectively suggests that systematic effort in manipulating gangliosides via UGCG or GD3 synthesizing enzymes can serve as a strategy to prevent the activation of multiple RTKs that network for accelerated tumor growth in the luminal subtype.

Targeting RICTOR downstream of RTKs may be another node that can be tapped simultaneously. We ruled out the cross-talk of mTORC1 on RICTOR silencing, as RAPTOR and downstream targets like pS6 Kinase and p4EBP1 were unchanged in RICTOR-silenced cells. However, many negative feedback loops are working between mTOR cascades, which, though important, investigating all of these was beyond the scope of this study. Although many studies have reported high UGCG expression in ER+ luminal tumors [51,52], none delved into its molecular mechanism. Our study, on the other hand, unraveled and validated that ZFX, a key C2H2-type, ZNF family transcription factor, mediates UGCG expression and thereby modulates sphingolipid and ganglioside metabolism and tumor progression. Thus, to avoid the activation of mTOR-mediated negative feedback loops promoting cell proliferation, efforts need to be diverted to the manipulation of ZFX-controlled *UGCG* expression that will mimic the effect of RICTOR inhibition without associated side effects. This regulatory effect of ZFX may be one of the reasons why genetic manipulations to silence ZFX in breast, colorectal, pancreatic, and renal cancers reduced proliferation and predicted good prognosis [39,53]. It also needs to

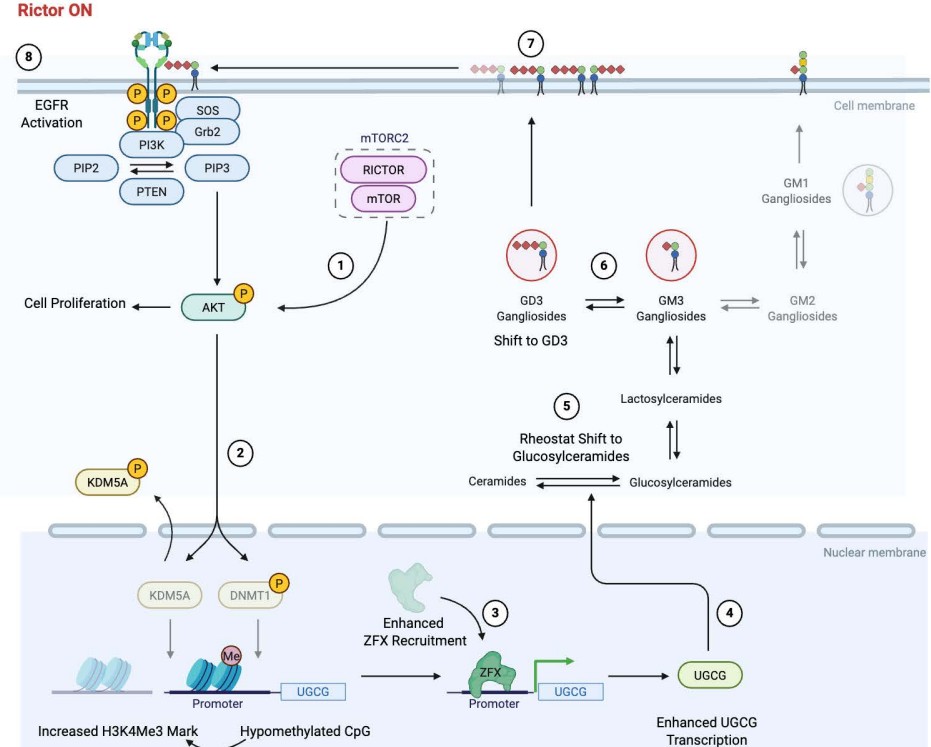

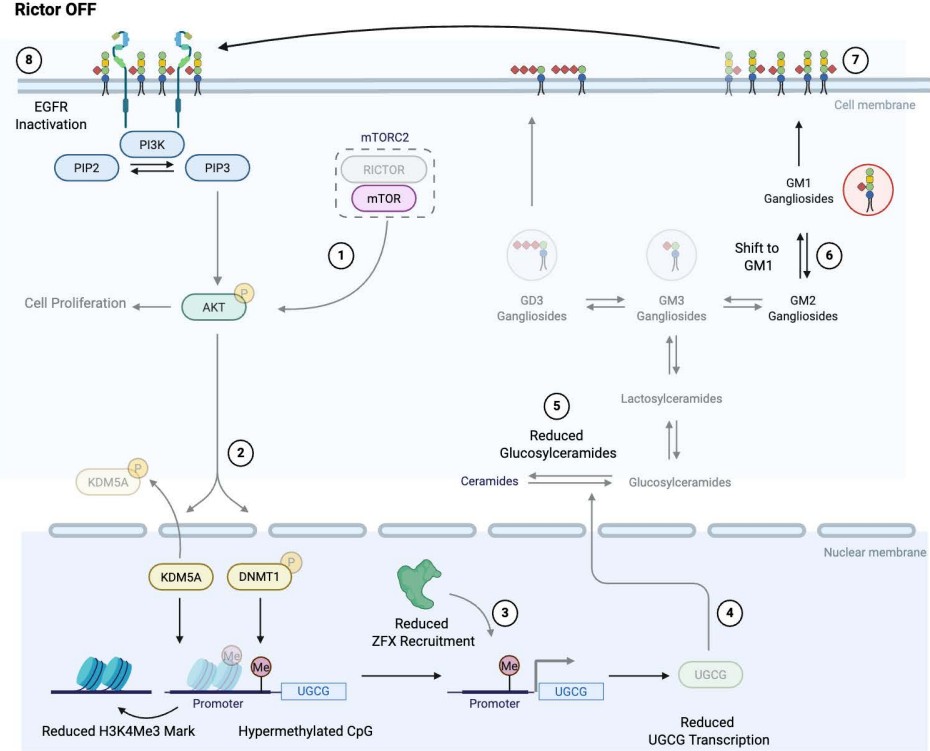

**Fig 7. Schematic showing the metabolite-cell signaling-gene regulatory circuit connecting mTORC2/RICTOR signaling to ganglioside metabolism, regulating uncontrolled breast cancer cell proliferation.** Stages 1–8 sequentially represent the mTORC2/RICTOR signaling-mediated epigenetic regulations modulating UGCG transcription, which leads to ganglioside-mediated EGFR activation and tumor progression. The solid black lines represent increased activation and signaling in the circuit, whereas gray lines represent attenuated signaling. Proteins/metabolites in solid colors are upregulated, and blurred colors are downregulated.

be emphasized that, depending on the cell line and growth signaling pathway involved, multiple transcription factor/s may regulate *UGCG* in a context-dependent manner. Although our study is focused on luminal breast cancer, we have shown that ZFX also regulates UGCG expression in other cancer cell lines. Further, we showed that RICTOR regulates three transcription factors, like ZFX, CTCF, and ELF1, that can further regulate multiple genes/enzymes, including some from the sphingolipid and ganglioside pathways in a cell-type and cancer-context-dependent manner. Therefore, there is a need to understand these regulatory mechanisms and their signaling targets in a cell and cancer-context-dependent manner.

Since the UGCG-mediated regulation of ganglioside metabolism is a key driver of proliferation, cells have evolved a two-pronged strategy driven by AKT to modulate promoter occupancy by ZFX [54]. In a cancer cell, activated AKT phosphorylates both KDM5A, leading to its nuclear exclusion, and DNMT, leading to its inactivation. While nuclear exclusion of KDM5A increases the H3K4Me3, a transcriptional activation mark, depletion of active DNMT leads to hypomethylation of promoters, both helping to increase ZFX access to chromatin and transcription of *UGCG*. Since RICTOR is known to regulate AKT phosphorylation [55], ZFX recruitment is reduced, and *UGCG* transcription is downregulated in RICTOR-silenced cells. In summary, our study endorses that the EGFR-mTORC2-RICTOR-AKT-UGCG-Ganglioside circuit regulates tumor progression in luminal breast cancer cells and provides us with an opportunity to intervene at multiple nodes to tame cancer cells. It is prudent to mention that the underlined paradigm of mTORC2/RICTOR-regulated expression of *UGCG,* impacting the level of gangliosides, may be part of a fundamental mechanism in breast tumor development, especially in luminal tumors, as shown in clinical samples, which certainly requires more attention and research. Recent studies have shown that gangliosides act as potential antigens and have a key role in immunosuppression. Therefore, one of the limitations of this study is to explore the effect of these RICTOR-mediated changes in gangliosides on the tumor microenvironment, which can be studied in syngeneic or humanized tumor models in the future.

## Materials and methods

### Materials

**Cell culture.** MCF-7, BT-474, MDA-MB-453, HCT-116, HEK-293 cells (ATCC, USA), DMEM media (Cat# D5648) Sigma, USA, MEBM media (Cat# CC-3151) Lonza, Switzerland, MEM media (Cat# AL081) HiMedia, USA, DPBS (Cat# D5652) Sigma, USA, FBS (Cat# 10270) Gibco, USA, lipid-free FBS (Cat# S148L) Biowest, USA, Penicillin-Streptomycin (Cat# 113-98-43810-74-0) HyClone, USA, Lipofectamine 2000, (Cat# 11668019) Invitrogen, USA, Lipofectamine 3000 (Cat# L3000015) Invitrogen, USA, Trypsin (Cat# TCL007) HiMedia, USA, Puromycin (Cat# P7255) Sigma, USA, G418 (Cat# A1720) Sigma, USA, Haemocytometer (Cat# Z359629) Bright-Line, USA, shRNA Control (Cat# SHC202V) Sigma, USA, shRNA RICTOR Virus Particles (Cat# SHCLNV, TRCN0000289691, TRCN0000307119, TRCN0000296313, TRCN0000307122) Sigma, USA, shRNA RICTOR Glycerol stocks (SHCLNG, TRCN0000296313, TRCN0000307122,), Sigma, USA, shRNA UGCG Glycerol stocks (SHCLNG, TRCN0000036128, TRCN0000036126, TRCN0000300623) Sigma, USA, ZFX Glycerol stocks (Cat# SHCLNG TRCN0000017308, TRCN0000017309, TRCN0000017310), Sigma, USA, shRNA ST8SIA1 (Cat# SHCLNG TRCN0000417447, TRCN0000036044, TRCN0000036046), Sigma, USA, UGCG siRNA (Cat# AM51331) Ambion, USA, ZFX siRNA (Cat# L-006572-00-0005) Dharmacon, USA, Scrambled siRNA (Cat# D-001810-10-05) Dharmacon, USA, GD3 Synthase (ST8SIA1) siRNA (Cat# EHU025731-20UG) Merck, USA, 5-Aza-2′-deoxycytidine (DAC) (Cat#A3656-10MG) Sigma, USA, KDOAM25 Hydrochloride hydrate (Cat# SML2774-5MG) Sigma,

USA, MK-2206 dihydrochloride (Cat# HY-10358) MedChem Express, USA, Eliglustat (Cat# HY-14885) MedChem Express, USA, Hygromycin (Cat# PCT1503), HiMedia, USA, Puromycin (Cat# P8833), Sigma, USA.

**Biochemicals and kits.** Qubit RNA HS Assay Kit (Cat# Q32853) Invitrogen, USA, RNAiso Plus (Cat# 9109) DSS Takara, India, RNeasy Lipid Tissue Mini Kit (Cat# 74804) Qiagen, Germany, Ethanol (Cat# 100983) Merck, USA, MOPS, free acid (Cat# MB0360) Bio basic, Canada, Formaldehyde (Cat# MB059) HiMedia, USA, Ethidium bromide (Cat# E8751) Sigma, USA, 100 bp DNA ladder (Cat# BM001-R500) BR Biochem, China, TURBO DNase (Cat# AM2238) Invitrogen, USA, iScript cDNA synthesis kit (Cat# 1708891) Bio-Rad, USA, Agarose (Cat# A9539) Sigma, USA, iTaq universal SYBR Green supermix (Cat# 1725124) Bio-Rad, USA, Ethylenediaminetetraacetic acid disodium salt dihydrate (Cat# E5134) Sigma, USA, Tris (Cat#MB029) HiMedia, USA, NaCl (Cat#GRM853) HiMedia, USA, $MgCl_2$ (Cat# 208337) Sigma, USA, $CaCl_2$ (Cat# 449709) Sigma, USA, Triton X-100 (Cat# T8787) Sigma, USA, Sodium deoxycholate (Cat# 1.06504) Millipore, USA, DTT (Cat# DTT-RO) Roche, Switzerland, complete Protease Inhibitor Cocktail (Cat# CO-RO) Roche, Switzerland, PhosSTOP (Cat# 04906845001) Roche, Germany, SUPERase In RNase Inhibitor (Cat# AM2694) Thermo Scientific, USA, Xylene cyanol FF (Cat# X4126) Sigma, USA, Hydrochloric acid (Cat# 29505) Thermo Fisher, USA, Bromophenol blue sodium salt (Cat# B8026) Sigma, USA, Sodium dodecyl sulfate (Cat# L3771) Sigma, USA, Lithium Chloride (Cat# 9650-100G) Sigma, USA, Acrylamide (Cat# AB1032) Bio basic, Canada, Polyoxyethylenesorbitan monolaurate (Tween 80) (Cat# GRM156) HiMedia, USA, Polyoxyethylenesorbitan monolaurate (Tween 20) (Cat# P7949) HiMedia, USA, Glycerol (Cat# GRM1027) HiMedia, USA, Ammonium persulfate (Cat# A3678) Sigma, USA, Phenylmethanesulfonyl fluoride (Cat# P7626) Sigma, USA, 1× Protease inhibitor cocktail (Cat# R1329) Fermentas, USA, HindIII (Cat# R0104S) NEB, USA, Xhol (Cat# R0146S) NEB, USA, NEB Buffer 2.1 (Cat# B7202S) NEB, USA, *N,N′*-methylene bisacrylamide (Cat# M7279) Sigma, USA, Sodium chloride (Cat# GRM853), HiMedia, USA, Glycine (Cat# G8898), Sigma, USA, Bovine serum albumin fraction-V, HiMedia, USA, nitrocellulose (Cat# GRM105), Merck, USA, PVDF membrane (Cat# IPVH00010), Merck, USA, Chemiluminescent HRP substrate (Cat# WBKLS0500), Merck, USA, Pierce BCA protein assay kit (Cat# 23227), Thermo Scientific, USA, MTT (Cat# M5655), Sigma, USA, Hoechst 33258 (Cat# 861405), Sigma, USA, Paraformaldehyde (Cat# 81847), Thomas baker, India, Cryomatrix (Cat# 6769006), Thermo Scientific, USA, Poly-lysine slides (Cat# P0425), Sigma, USA, Goat serum (Cat# RM10701), HiMedia, USA, Allprotect Tissue Reagent (Cat# 76405), Qiagen, Germany, Prolong gold anti-fade reagent (Cat# P36934), Life technologies, USA, Sectioning blade (Cat# 152580), Micron, India, Triton X-100 (Cat# T9284), Sigma, USA, Taurocholic acid sodium (Cat# T4009), Sigma, USA, Citric acid (Cat# 251275), Sigma, USA, Disodium hydrogen orthophosphate dihydrate (Cat# 40158), S.D. Fine, India, Sodium hydroxide (Cat# 40167), S.D. Fine, India, Kanamycin (Cat# 25389-94-0), GoldBiocom, India, Dynabead Protein A (Cat# 10002D), Thermo, USA, Dynabead Protein G (Cat# 10004D), Thermo, USA, Poly(ethylene glycol) 8000 (Cat#P2139-1KG) Sigma, USA, Protein A/G PLUS-Agarose beads (Cat# sc-2003) Santa Cruz, USA, Plasmid Midi Prep (Cat# 12143), Qiagen, Germany, Polybrene transfection reagent (Cat# T1003), Sigma, USA, Developer (Cat# 4908216), Carestream, USA, Protein Ladder (Cat# 26616, Cat# BP3603500) ThermoFisher, USA, Protein Ladder (Cat# PG-PMT2922), Genetix, India, Fixer (Cat# 4908232), Carestream, USA, XBT X-Ray film (Cat# 6568307), Carestream, USA, Immobilon Western Chemiluminescent HRP (Cat# WBKLS0500), Merck Millipore, USA.

**Chemicals for lipidomics and mass spectrometry studies.** Methanol (MS Grade Cat#34966), Honeywell, USA, Chloroform (MS Grade Cat#25669-1L), Honeywell, USA, 2-Propanol (Cat#34965), Honeywell, USA, Formic Acid (MS Grade Cat# 56302-50ML), Fluka, USA, Ammonium formate, (Cat# 14266-25G), Honeywell Fluka, USA, Acetonitrile (Cat#34967) Honeywell, USA, Ammonium acetate (Cat#14267-25G), Honeywell, USA, Ammonium hydroxide (Cat# 16227), Thermo Fisher, USA, Triethyl ammonium bicarbonate buffer Sodium Hydroxide (Cat# T70408), Sigma, USA, Sodium Hydroxide (Cat# 13913) SRL Chem, India, Chymotrypsin (Cat# 11418467001), Merck, USA, Water (MS Grade Cat#39253-4L) Riedel-de haen, Germany. Potassium hydroxide (Cat#84749), Sisco Research, India, Iodoacetamide (Cat# 144-48-9) Sigma, USA, Glacial acetic acid (Cat# 144-48-9) Merck, USA, Digitonin (Cat# D141) Sigma, USA, ACQUITY UPLC BEH Shield RP18 column

(Cat#186002854) Water Ltd., ChromXP C18-CL trap column (Cat#5016752) was purchased from Eksigent, ABSciex, USA, nanoViper C18 column (Cat# 164569) Thermo Scientific, USA, Kinetex C18 column (Cat#00B-4601-AN) Phenomenex, USA, Ceramide/Sphingoid Internal Standard Mixture II (Cat# LM6005-1EA) Avanti Polar Lipids, USA, C18 Ganglioside GM3-d3 (d18:1/18:0d3) (ammonium salt) (Cat# 24850) Cayman Chemicals, USA, Ganglioside GM3 (Bovine Milk) (Cat# 860058P) Avanti Polar Lipids, USA, Ganglioside GM1 (Bovine Brain) (Cat# 860065P) Avanti Polar Lipids, USA, Ganglioside GD3 (Bovine Milk) (Cat# 860060P) Avanti Polar Lipids, USA, FITC labeled cholera toxin B (Cat#1,655) Sigma-Aldrich, USA.

**Antibodies.** Ceramide Glucosyl transferase (Cat#ab124296) Abcam, UK, Ceramide Glucosyl transferase IHC (Cat# ab197369) Abcam, UK, Ceramide Glucosyl transferase (Cat# 12869-1-AP) ProteinTech, USA, GBA1 (Cat# ab88300) Abcam, UK, eIF4EBP1 (Cat# ab32024) Abcam, UK, P-eIF4EBP1 (Cat# ab75767) Abcam, UK, DNMT1 (Cat# ab188453) Abcam, UK, DNMT1 (Cat# MA5-16169) Invitrogen, USA, H3K4ME3 (Cat# ab1791) Abcam, UK, H3 (Cat# ab8580) Abcam, UK, Rictor (Cat# 2140S) Cell Signaling, USA, Raptor (Cat# 2280) Cell Signaling, USA, Akt (Cat# 4685) Cell Signaling, USA, P- Akt (Cat# 4060) Cell Signaling, USA, P-SGK1 (Cat# 5599) Cell Signaling, USA, P70 S6 Kinase (Cat# 5599) Cell Signaling, USA, ZFX (Cat# 5419) Cell Signaling, USA, ZFX (Cat# PA5-78234) Abcam, UK, JARID1A (KDM5A) (Cat# 3876S) Cell Signaling, USA, KDM5A (Cat# ab70892), Abcam, UK, Phosphoserine (Cat# ab9332) Abcam, UK, EGFR (Cat# 2232) Cell Signaling, USA, P-EGFR (1068) (Cat# 2232) Cell Signaling, USA, P-EGFR(1178) (Cat# 2234) Cell Signaling, USA, Erk1/2 (Cat# 9102S) Cell Signaling, USA, P- Erk1/2 (Cat# 9101) Cell Signaling, USA, Anti-ceramide antibody (Cat# C8104) Sigma, USA, GM3 Synthase (ST8SIA-V) (Cat# sc-365329) Santa Cruz, USA, B3GALT4 (Cat# ab169759) Abcam, UK, GM2/GD2 Synthase (B4GALNT1) (Cat# sc-376505) Santa Cruz, USA, NEU1 (Cat# 67032-1-Ig) ProteinTech, USA, NEU2 (Cat# 24523-1-AP), ProteinTech, USA, NEU3 (Cat# 27879-I-AP), ProteinTech, USA, NEU4 (Cat# 12995-I-AP) ProteinTech, USA, HEXA (Cat# 60337-I-Ig) ProteinTech, USA, HEXB (Cat# 16229-I-AP), ProteinTech, USA, GLA (Cat# 66121-I-Ig), ProteinTech, USA, GLB (Cat# 66586-I-Ig) ProteinTech, USA, β-actin, (Cat# A5441), Sigma, USA, β-actin (Cat# 66009-1-Ig), ProteinTech, USA, Secondary IgG anti-mouse Alexa fluor 594 (Cat# 88903) Cell Signaling, USA, FITC anti-mouse IgM (Cat# 406506) BioLegend, USA, Goat anti-mouse IgG (L+H) (Cat#ab6789) Abcam, UK, Goat anti-rabbit IgG-HRP (Cat# sc-2004) Santa Cruz, USA, Rabbit IgG isotype control (Cat# 02-6102), Invitrogen, USA, Mouse IgG2a Isotype Control (Cat# 02-6200) Invitrogen, USA.

## Methods

**Cell culture.** Human cell lines MCF-7, BT-474, HCT-116, HepG2, MDA-MB-453, and HEK-293 obtained from American Type Culture Collection (ATCC, Manassas, VA, USA) were cultured in DMEM media with 10% Fetal bovine serum, 100 units/mL penicillin, and 100 μg/mL streptomycin. Cells were grown at 37 °C with 5% $CO_2$ in a humidified incubator.

**Cloning.** UGCG and ZFX genes were cloned in pBBL-FLAG vector (BioBharti LifeScience Pvt Ltd) using Xho1 (NEB, R0104S) and Hind III (NEB, R0146S) restriction enzymes. UGCG, UGCG-7 (Dead), ZFX, ST8SIA1, and B3GALT4 genes were cloned in pLVX-IRIS-Hygro (NEB, R0145S) using XbaI and XhoI restriction enzymes. For amplification, the recombinant plasmids were transformed into *Escherichia coli* (DH5α) competent cells, and the positive colonies were selected using kanamycin. The plasmids were extracted using the plasmid extraction kit.

## Engineering of cell lines

MCF-7 and BT-474 cells were transfected with plasmid DNA (UGCG or ZFX or empty vector) or siRNA (targeting UGCG or ZFX or scrambled) using Lipofectamine 2000. About $3.5 \times 10^5$ MCF-7/BT-474 cells per well were seeded in a six-well plate in DMEM-high glucose with 10% FBS and 10% penicillin and streptomycin, and incubated at 37 °C in $CO_2$ incubator for 24 h. At 80%–85% confluency, plasmid and Lipofectamine complex in a 1:3 ratio were incubated for 25 min in Mammary Essential Medium Eagle, and cells were transfected with these complexes. After 6 h of transfection, antibiotic-free media containing 10% FBS were added to the cells, and cells were incubated for 36 h. For the generation of stable cell lines, transfected cells were selected with a G418 antibiotic-resistant marker.

For generating lentiviral-mediated overexpression cell lines, viral particles were produced by co-transfecting HEK293T cells with a lentiviral expression plasmid vector pLVX-IRIS-Hygro (Cat# 632185, Takara) and pLVX-IRIS-Hygro cloned with UGCG, UGCG-7 (Dead), ZFX, ST8SIA1, B3GALT4 along with packaging plasmids pMDLg/pRRE (Cat# 12251, Addgene), PMD2.G/VSVG (Cat# 12259, Addgene), and PRSV/REV (Cat# 12253 Addgene) in a ratio of 4:4:1:1 using Lipofectamine 3000 (Invitrogen, L3000015) [56]. For knockdown cell lines, viral particles were similarly generated using a lentiviral shRNA expression plasmid vector pLKO.1-puro (Cat# 8453, Addgene) [57] and shRNA of RICTOR, UGCG, ZFX, and ST8SIA1 (obtained from Sigma) with the same packaging plasmids and transfection conditions. Viral supernatants were collected 72 h post-transfection, filtered through a 0.45 µm syringe filter, and concentrated using a Lenti-X concentrator. Target cells were transduced in the presence of 4 µg/mL polybrene (Merck, TR-1003-G). Transduced cells were selected using hygromycin for overexpression or puromycin for knockdown.

## Collection of patient tumor tissue

Patient tumor and adjacent normal breast tissue were collected from operable luminal breast cancer patients (Stages I, II, IIIa) undergoing treatment at Dr. B. R. Ambedkar Institute-Rotary Cancer Hospital (BRA-IRCH), All India Institute of Medical Sciences (AIIMS), New Delhi, and from biorepository of Rajiv Gandhi Cancer Institute and Research Center (RGCIRC), Delhi, after due ethical approvals. Informed consent was taken from all the patients before acquiring samples. Inclusion criteria included women patients of all ages (18–85 years) and socioeconomic status who gave consent for tissue collection, patients with operable breast cancers (Stages I, II, IIIa) who will undergo adjuvant therapy, and patients with ER$^+$PR$^+$HER2$^-$ status. Exclusion criteria included patients who were undergoing neo-adjuvant chemotherapy and patients who were not fit to undergo surgery. Details of patients and tumor signatures are mentioned in S1 Table.

## Cellular assays

For siRNA silencing, cells (~$3.5 \times 10^5$ per well) were seeded in six-well plates containing DMEM supplemented with 10% FBS and 10% penicillin-streptomycin, then incubated at 37 °C with 5% $CO_2$ for 24 h. Once cells reached 80%–85% confluency, siRNA targeting UGCG, ZFX, ST8SIA1, or scrambled control was transfected using Lipofectamine 2000. siRNA and Lipofectamine were mixed at a 1:3 ratio in MEM media and incubated for 25 min before adding to the cells. After 6 h of transfection, the media were replaced with antibiotic-free DMEM containing 10% FBS, and cells were incubated for an additional 36 h. Silencing efficiency was subsequently confirmed by western blotting.

For inhibitor experiments, cells were seeded in six-well plates with DMEM supplemented with 10% FBS and 10% penicillin-streptomycin and incubated at 37 °C with 5% $CO_2$ for 24 h. Once cells reached 40%–60% confluency, cells were treated with 5 µM DAC for DNMT1 inhibition, 30 µM KDOAM-25 for KDM5A inhibition, and 4 µM of MK2206 for pAKT inhibition. All treatments were carried out for 48 h.

For proliferation assay, stable cell lines or siRNA-transfected cells (5,000 cells/well) were seeded in a 96-well plate for 24, 48, and 72 h in complete DMEM media and incubated at 37 °C. Cell proliferation was quantified using the MTT assay at 570 nm following the previously described method [58]. To assess the effect of gangliosides on cell proliferation upon RICTOR silencing in MCF-7 and BT-474 cells, 2,000 cells/well of MCF-7SCRAM$^{SH}$ and MCF-7_RICTOR$^{SH}$ and 1,000 cells/well of BT-474 SCRAM$^{SH}$ and BT474_RICTOR$^{SH}$ were seeded in a 96-well plate in complete DMEM media and incubated at 37 °C. After 24 h, cells were treated with either DMSO or glucosylceramides, and gangliosides including GM1, GM2, GM3, GD2, and GD3 in DMEM media supplemented with 10% lipid-free serum. Cells were then incubated for 96 h, and cell proliferation was quantified using MTT assay [58].

## Pellet collection for RNA and protein isolation

For RNA isolation, cells were grown in 100 mm cell culture plates. For pellet collection, the media was aspirated from the plates and washed two times with DPBS. 1 mL of Trizol was added to the plate, and incubated for 5 min. After incubation, cells were

scraped and transferred into 1.5 mL centrifuge tubes, and used immediately or stored at −80 °C. For protein isolation, cells were similarly washed, scraped out in DPBS, centrifuged at 5,000 rpm for 5 min, and used immediately or stored at −80 °C.

## Quantitative real-time PCR

Total RNA was extracted using an already standardized protocol [58]. The concentration of the RNA was determined using Nanodrop 2000 (ThermoScientific). The integrity of the RNA was checked on a 1% agarose gel. cDNA synthesis and real-time PCR were done as described previously [58]. Relative quantitation of gene expression was done using $\beta$−actin as the endogenous reference gene for normalization. All primer sequences used for RT PCR are listed in S3 Table.

## Western blotting

Protein expression analysis was done by western blotting as per the previously described method [58]. Protein separation was done on 10–12% SDS-PAGE with 25–60 μg of protein. After separation, proteins were transferred to the PVDF/Nitrocellulose membrane. Immunostaining was done by overnight incubation of the blot with the corresponding primary antibodies at 4 °C in 5% BSA/Skimmed milk in TBST. After washing, the blots were incubated with a secondary antibody for 1 h, and X-ray sheets were developed using Immobilon Western Chemiluminescent HRP.

## Immunoprecipitation

For immunoprecipitation experiments, cells were collected and lysed in immunoprecipitation buffer (25 mM Tris-HCl, pH 7.5, 150 mM NaCl, 1 μM EDTA, 5% Glycerol, 1% NP40) with phosphatase inhibitor and protease inhibitor cocktail. After protein estimation, lysate was precleared and incubated with mouse/rabbit IgG, DNMT1, and KDM5A antibodies (1:100) overnight at 4 °C with rotation. 20 μL of protein A/G agarose beads were added, followed by 2–4 h of incubation with rotation at 4 °C. After 2–3 washes with IP buffer, protein complexes were boiled in 40 μl of 2× SDS-PAGE sample buffer (10% glycerol, 62.5 mM Tris HCl pH 6.8, 2% SDS, 0.01 mg/mL bromophenol blue, 5% β-mercaptoethanol) and resolved on SDS-PAGE followed by immunoblotting.

## Cytoplasmic and nuclear extract preparation

After three washes with PBS, ~10 million cells were collected. For cytoplasmic extract, cell pellets were lysed in CEB (cytoplasmic extraction buffer: 10 mM HEPES, 3 mM $MgCl_2$, 20 mM KCl, 5% glycerol, 0.5 mM DTT, 0.5% NP40, protease inhibitors, and phosphatase inhibitors) for 30 min on ice and centrifuged at 1,300$g$ for 15 min. After removing the cytoplasmic extract, pellets were washed 3–4 times with CEB and lysed with NEB (nuclear extract buffer: 20 mM HEPES, 3 mM $MgCl_2$, 225 mM NaCl, 1 mM EDTA, 10% glycerol, 0.5 mM DTT, 0.5% NP40, protease inhibitors, and phosphatase inhibitors) for 30 min on ice followed by centrifugation at 13000$g$ for 15 min.

## Bioinformatic analysis

Differential gene expression analysis was performed on mouse rictor (−) microarray datasets (GSE46515, GSE67077, GSE84505, GSE85555) obtained from the NCBI GEO dataset using the limma package in R [59–63]. To study the effect of *RICTOR* knockdown on the transcription factors, the downregulated genes were compared with a list of mouse transcription factors from the Animal TFDB 3.0 database and categorized accordingly [64]. Finally, to study whether RICTOR knockdown had any effect on the transcription factors that bind to the UGCG promoter, the list of transcription factors was compared with a list of experimentally validated transcription factors binding to the UGCG promoter region (−3kb upstream of ATG) using the TRANSFAC software [65]. From this analysis, three potential transcription factors that may bind to the UGCG promoter were identified.

## ChIP qRT-PCR primer designing

For designing the ZFX and H3K4ME3 ChIP primers, we used ChIPBase v2.0 database (http://rna.sysu.edu.cn/chipbase/). For ZFX primers, we selected the transcription factor in the factor type option and looked for the binding of the ZFX transcription factor on the UGCG promoter in the DAOY medulloblastoma cell line. From the genome of DAOY medulloblastoma cell line, we have selected the ZFX binding region (coordinates; 111896151-111896592) on UGCG promoter provided by ChIPBase v2.0 for designing the ZFX ChIP primer. For H3K4Me3 ChIP primers, we selected the histone modification in factor type option, and looked for binding of H3K4ME3 on UGCG promoter, and selected the region of UGCG DNA (coordinates; 111896051- 111898139) for H3K4Me3 ChIP primer. Primer sequences for ChIP PCR are given in S4 Table.

## Chromatin immunoprecipitation (ChIP)

Cells were grown in a 100 mm dish, and on 80%–90% confluency (10–15 million cells), cross-linking of the proteins in cells was done using 1% formaldehyde. Formaldehyde was directly added to the dish, followed by incubation for 10 min at room temperature, and the reaction was quenched by adding 1/8 volume of 1M glycine and incubating for 5 min. After washing with PBS, cells were scraped in PBS and resuspended in 1.5 mL nuclear lysis buffer (1% SDS, 10 mM EDTA, 50 mM Tris-HCl pH-8.0, 1× protease inhibitor cocktail). Cells were then sonicated using Bioruptor (Diagenode, Denville, New Jersey, USA) for 38–44 cycles, keeping maximum amplitude with 30 s pulse and 30 s hold. Chromatin shearing was checked on a 1% agarose gel to confirm the correct size (approximately 300–500 bp) of chromatin. Protein estimation was done using a Bicinchoninic acid (BCA) protein estimation kit according to the manufacturer's protocol. Preclearing was done by incubating the protein with A/G magnetic beads for 1 h at 4 °C while rotating. For immunoprecipitation, 500 μg of the chromatin was incubated with 2.5 μg of antibody (ZFX, H3, H3K4Me3, or IgG) overnight at 4 °C while rotating. The next day, 50 μL of protein A/G magnetic beads were added and incubated for 2 h. Each sample was then subjected to one wash with 1 mL of low salt buffer (0.1% SDS w/v, 1% Triton X 100, 2 mM EDTA pH 8.0, 150 mM NaCl), high salt buffer (0.1% w/v SDS, 1% Triton X 100, 2 mM EDTA pH 8.0, 500 mM NaCl), LiCl buffer (20 mM Tris HCl, 500 nM NaCl, 2 mM EDTA pH 8.0, 0.1% w/v SDS, v/v 1% IGEPAL), and finally three washes with TE buffer (10 mM Tris HCl pH 8.0, 1 mM EDTA pH 8.0). All washes were incubated for 5 min at 4 °C while rotating. Next, 500 μL elution buffer (1M NaHCO$_3$, 10% SDS) was added for elution of chromatin from the antibody, followed by incubation for 5 min. Reverse crosslinking was done by adding 4 μL of 10% SDS (final concentration, 0.2%) and incubating at 65 °C overnight in a mixer with constant agitation at 1,200 rpm. RNase treatment was done for all the samples (including the input samples) by adding RNase A (100 μg/mL) to a final concentration of 0.2 μg/μL, followed by incubation for 2 h at 37 °C. Proteinase K (10 mg/ml) was added to the samples to a final concentration of 200 μg/mL, followed by incubation for 2 h. The DNA isolation was done by the phenol/chloroform/isoamyl alcohol (25:24:1) method. The DNA pellets were air-dried and dissolved in 20 μL of 10 mM Tris buffer (pH 8.0). Final DNA estimation was done using a Qubit dsDNA HS assay kit. ChIP-qPCR was carried out with an equal amount of ChIP DNA per reaction.

## ChIP RT-PCR

For ZFX ChIP RT-PCR, the ChIP DNA was quantified using Qubit HS DNA Kit, and ChIP-qPCR was performed with an equal amount of DNA from ChIP, Input, and its IgG control DNA. The mean Ct value was used to calculate fold change after normalization with IgG and Input. For H3K4ME3, the normalization was done using input and total H3. For ZFX peak calling, the ENCODE (PMID: 22955616; PMCID: PMC3439153, PMID: 29126249; PMCID: PMC5753278) database was used. The ENCODE identifier used was ENCSR435OQD (ZFX). For both transcription factors, the peaks were visualized using the UCSC genome browser.

## EMSA

To capture ZFX binding to UGCG DNA, we performed an Electrophoretic mobility shift assay (EMSA) adhering to the previously published protocol [66]. Briefly, we prepared nuclear extracts from MCF-7 and BT-474 cells overexpressing ZFX using the high-salt method. Cells transfected with the vector alone were used as controls. Approximately 1 mg of nuclear

extracts were incubated with [32]P-labeled double-stranded oligonucleotides (see below the sequence), whose sequence was derived from the UGCG promoter based on the ChIP-Seq analyses, for 20 min at room temperature. Subsequently, DNA-binding complexes were resolved on a native gel. Furthermore, a shift-ablation assay was performed to verify the identity of specific ZFX-DNA complexes. Here, nuclear extracts were incubated with ZFX (a cocktail of ZFX antibodies) prior to adding the labeled probe (lanes 4). The specificity of DNA binding was also tested in a competition assay, where nuclear extracts were first incubated with a 20-fold molar excess of unlabeled UGCG probe for 20 min prior to incubating with radiolabelled DNA (lane 5). A 20-fold excess of unrelated oligo was used in the competition assay as a control (lane 6). "*" indicates nonspecific complexes. UGCG oligo used were: GGA AGC CCG GCC TGC GTC CTG CGG; CCG CAG GAC GCA GGC CGG GCT TCC.

## Isolation and quantification of sphingolipids using LC–MS/MS

Collection of cell pellets, lipid isolation, LC–MS/MS analysis, and absolute quantitation of sphingolipids were performed as per the published protocol [58].

## Isolation and quantification of gangliosides using LC–MS/MS

The cell pellets or tissues were resuspended in LC–MS grade water and homogenized using bead ruptor and probe sonicator. An aliquot was taken for protein estimation by the BCA protein estimation kit. The cell suspension with 500 μL of water was transferred to Teflon-lined borosilicate tubes and mixed with deuterated, GM3-d3 (d18:1/18:0-d3), GD3-d3 (d18:1/18:0-d3), and GM1-d3 (d18:1/18:0-d3) internal standards. Chloroform: methanol (2:1) (4 mL) was added to the solution, placed overnight at 4 °C, and centrifuged. The supernatant was transferred to another tube, and the pellet was extracted again. The solution was then mixed with 1.3 mL of water to get separation, and the upper phase was transferred to another tube. The lower phase was mixed with 1 mL of water and centrifuged. The upper layer was added to the previous tube and dried under nitrogen gas to get a 2 mL volume. For enrichment of gangliosides, samples were passed through a C18 Sep-Pak cartridge. The Sep-Pak column was washed with 2 mL of chloroform: methanol (1:1), followed by 2 mL of methanol, and finally equilibrated with 2 mL of water. The samples were passed through the Sep-Pak column and washed with 0.5 mL of water. The elution was done using 2 mL of methanol and evaporated under N2 gas. Dried samples were resuspended in 200 μL of solvent B [methanol: isopropanol (1:1) with 0.2% formic acid and 5 mM ammonium acetate)]. The samples were vortexed, centrifuged, and transferred to the autoinjector vial for LC–MS/MS analysis using high-pressure UHPLC liquid chromatography (Exion LC AC, SCIEX, USA) coupled to a hybrid triple quadrupole/ linear ion trap mass spectrometer (6500+ QTRAP, SCIEX, USA) using multiple reaction monitoring (MRM). A Kinetex C18, 2.1 × 50 mm column (Phenomenex) with a particle size of 1.7 μm was used at an oven temperature of 40 °C. Total optimized run time was 32 min where solvent A (0.2% formic acid and 5 mM ammonium acetate in water) and solvent B [methanol:isopropanol (3:1) with 0.2% formic acid and 5 mM ammonium acetate)] were used as mobile phase A and B with flow rate of 0.2 mL/min. The gradient started with 60% solvent B. At 3.5 min, it changed to 80% of solvent B. By 18.5 min, it further increased to 90%. At 18.6 min, it reached 100% and maintained this until 28.6 min. Subsequently, at 29 min, it reverted back to 60% of B to retain the initial condition until 32 min. The analyte peak area for all MRM experiments was integrated into MultiQuant 3.0.2 for data analysis to quantify all ganglioside species in samples. All the parameters for the estimation of gangliosides are mentioned in S5 Table.

## Animal studies

All tumor growth kinetic studies were performed using MCF-7, BT-474 and their derivatised cell lines in NOD SCID C.B-17 mice. The flank region of the mice was shaved with Veet to remove the hair before the cell injection. The cells, suspended in FBS: Matrigel (1:1, 200 μL), were injected (1.5 × 10[6]) subcutaneously in the flank region. Once the tumor volume

reached 40–50 mm$^3$ after 7–8 days, measurement of tumor volume was commenced every 2 days and was done every 2 days. Tumor volume was calculated using the formula L*B$^2$/2, where L is length and B is breadth. On the final day of the measurement, the tumor was harvested and stored in Allprotect tissue reagent at −80 °C.

For siRNA experiments, mice were randomized into different groups after the tumor reached 50 mm$^3$ volume. Group 1 mice were left untreated; group 2 mice were treated with scrambled siRNA, and group 3 mice were treated with target siRNA. Group 2 and 3 mice were treated with 50 µL of 200 ng of siRNA complexed with TAC6 polymer (siRNA: TAC6 polymer; 1:10) at the tumor site. A total of 6 doses were given at an interval of every 2 days [67]. At the end of the experiment, the tumor was excised and stored at −80 °C for further analysis.

For inhibitor experiments, mice were randomized into two groups after the tumor reached 50 mm$^3$ volume. Group 1 mice were treated with saline (vehicle control) every day. In group 2, mice were treated intraperitoneally with eliglustat dissolved in saline at 60 mg/kg daily. Tumor volume was measured every day for different groups.

## Immunofluorescence in tissue sections

Patient tumor and adjacent normal tissues included in Supplementary data set 1, frozen at −80 °C in Allprotect tissue reagent, were processed, fixed, and stained following the already described method [58]. The tissue sections were stained with RICTOR antibody. Confocal imaging of the samples was performed with a Leica TCS SP8 microscope. The sections were visualized at 40× oil immersion using LAS AF software. Z-stacking was performed, and images were acquired for each section. The images were processed using LAS X software.

## Immunohistochemistry (IHC) in human tissue samples

IHC for UGCG and ZFX was carried out on Tissue microarray sections of breast cancer using standard protocol after due ethical approval. Briefly, 5 µm FFPE tissue sections were fixed on glass slides, followed by deparaffinization in xylene, rehydration in graded alcohol, 3% H2O2 treatment for 30 min, antigen retrieval using citrate buffer (pH 6.00) for 15 min, and blocking with 3% bovine serum albumin for 30 min. After this, sections were incubated with primary antibody against UGCG (dilution 1:250) and ZFX (dilution 1:100) for 1 h at room temperature. Incubation with a secondary antibody (DAKO REALEnVision) was done for 30 min. 3−3′ Diaminobenzidine (DAB, DAKO REALTMEnVisionTM) for 10 min was used as a chromogenic substrate, and hematoxylin was used as a counterstain. Sections were dehydrated, dried, mounted with DPX, and visualized under a microscope. All sections were examined by a pathologist to score tumor cells. Sub-localization of the staining in the cytoplasm and nucleus was recorded separately. Clinical details of patients are provided in S2 Table.

## Analysis of the TCGA and METABRIC datasets

We accessed the TCGA dataset (https://www.cancer.gov/tcga). This dataset had RNA sequencing data of 1,082 breast cancer patients. Similarly, the METABRIC data set was accessed with 1905 tumors with gene expression levels available from microarray [47]. In both datasets, expression levels of UGCG and ZFX were compared between PAM 50 subtypes and by estrogen receptor status. 601 tumors were identified as Luminal A+B after eliminating other tumors within TCGA and 1,170 within the METABRIC datasets. The mean expression levels of UGCG and ZFX were used as a cutoff within the luminal tumors to divide the tumors as high and low for both genes, respectively. Expression levels of ZFX were compared between the high and low UGCG-expressing tumors within the luminal groups.

## scRNA analysis

Single-cell RNA sequencing (scRNA-seq) data were obtained from the Gene Expression Omnibus (GEO) under accession number GSE176078, which includes 26 primary breast tumors across three clinical subtypes [68]. For this study, we

analyzed only the 11 estrogen receptor-positive (ER+) tumors, profiled using the 10× Genomics Chromium platform. Raw gene-barcode matrices (genes.tsv, barcodes.tsv, and matrix.mtx) were processed in R using the Seurat (v5.2.0) package [69]. Quality control filtering was applied to remove cells with fewer than 200 detected genes or greater than 20% mitochondrial gene expression. Data were log-normalized and scaled for downstream analyses. Cell identities were assigned based on the celltype_major annotation provided in the metadata, which classified cells into categories such as cancer epithelial, T-cells, B-cells, myeloid cells, and endothelial cells. The expression patterns of RICTOR, ZFX, and UGCG were examined across these major cell types within ER+ tumors to evaluate their cell type-specific expression profiles.

### RNA isolation and quantitative real-time PCR from patient tumor tissues

RNA isolation and quantitative RT PCR from luminal A patient tissue samples (~20 mg) stored in Allprotect Tissue Reagent were performed using the reported method [58]. cDNA synthesis and real-time PCR were done as described above.

### Statistical analysis

Statistical analyses were carried out using GraphPad Prism 10 (GraphPad Software). All data are represented as mean ± SD or mean ± SEM. A minimum of three biological replicates were used per condition in each experiment, as mentioned in each figure legend. Pairwise comparisons were determined using Student $t$ test. Multiple comparisons among groups were determined using one-way ANOVA followed by a post-hoc test. Growth kinetic analysis was performed using Two-way ANOVA. Differences between groups were considered significant at $p$-values below 0.05 (*$p < 0.05$, **$p < 0.001$, *** $p < 0.0001$, and ****$p < 0.0001$).

### Ethical statement

All animal experiments were performed after due approval of the Institutional Animal Ethical Committee (IAEC) of the Regional Centre for Biotechnology (Approval No. RCB/IAEC/2020/079) as per the guidelines of the Committee for Purpose of Control and Supervision of Experiments on Animals (CPCSEA), India. All studies with human tissue samples were conducted after due ethical clearance from Institute Ethics Committee (Human) of All India Institute of Medical Sciences New Delhi (Approval No. IEC-332/01.07.2016), Institute Review Board of Rajiv Gandhi Cancer Institute and Research (Approval No. RGCIRC/IRB/276/2019, Res/BR/TRB-20/2020/70), Institute Ethics Committee (Human) of Amity University Haryana (Approval No. IEC-AIISH/AUH/ 2016-1), and Institute Ethics Committee (Human) of Regional Centre for Biotechnology (Approval No. RCB-IEC-H-09) and Institute Review Board of St John's Research Institute (Approval No. S475/79-80) after the due written consent from the patients.

### Supporting information

**S1 Fig. Related to** Fig 1. (**A, B**) Absolute quantitation (pmol/mg protein) (mean ± SEM, $n = 27$) of different species of ceramides (A) and glucosylceramides (B) shows higher levels in luminal tumor tissues (labeled as T) in comparison to adjacent normal tissues (labeled as N). (**C**) Immunoblot confirm increased UGCG expression in BT-474_UGCG^OE cells compared to BT-474 cells. (**D**) Quantification of glucosylceramides (mean ± SEM, $n = 4$) confirms an increase in BT-474_UGCG^OE cells over BT-474 cells. (**E–H**) Absolute quantification (mean ± SEM, $n = 3$–4) of GM3 (E), GD3 (F), GM2 (G), and GM1 (H) gangliosides shows an increase in GD3 and GM2 gangliosides and a decrease in GM3 and GM1 gangliosides in BT-474_UGCG^OE cells compared to BT-474 cells. (**I**) Cell proliferation (mean ± SEM, $n = 5$) assay demonstrates an increase in the proliferation of BT-474_UGCG^OE cells over BT-474_VECT^OE cells. (**J–M**) Immunoblot confirms silencing of UGCG in MCF-7 and BT-474 cells (J, L) and cell proliferation assay demonstrates a decrease in proliferation in MCF-7_UGCG^SL and BT-474_UGCG^SL cells over MCF-7_SCRAM^SL (mean ± SEM, $n = 5$) and BT-474_SCRAM^SL cells (mean ± SEM, $n = 3$) (K, M). (**N**) Tumor growth kinetics reveal enhanced growth of BT-474_UGCG^OE tumors compared to

BT-474_VECT$^{OE}$ tumors (mean ± SEM, $n = 5$). (**O**, **P**) Fold change of glucosylceramides (mean ± SEM, $n = 4$) confirms no significant increase in glucosylceramides in MCF-7_UGCG$^{DEAD}$ and BT-474_UGCG$^{DEAD}$ cells compared to MCF-7 and BT-474 cells. (**Q**) Immunoblots show the expression of RICTOR, RAPTOR, AKT, pAKT$^{Ser473}$, SGK1, pSGK1$^{Ser78}$, 4EBP1, p4EBP1$^{Thr37}$ and p70S6K in BT-474_UGCG$^{OE}$ cells in comparison to BT-474_UGCG$^{DEAD}$ cells. (**R**) Immunofluorescence images show elevated RICTOR expression in tumor tissues compared to adjacent normal tissue sections. Data among groups were analyzed using a paired Student $t$ test (for patient data), One-way ANOVA among multiple groups or Two-way ANOVA in time-dependent studies. *P*-value: \**p* < 0.05, \*\**p* < 0.01, \*\*\**p* < 0.0005, \*\*\*\**p* < 0.0001. Numerical data can be found in S1 Dataset.
(S1_Fig.TIF)

**S2 Fig. Related to** Fig 2. (**A**) Immunoblots confirm knockdown of RICTOR expression in BT-474_RICTORSH cells. (**B**) Immunoblots show changes in expression of RICTOR, RAPTOR, and their downstream effectors in BT-474_ RICTOR$^{SH}$ cells compared to BT-474_SCRAM$^{SH}$ cells. (**C**) Cell proliferation studies show a decrease in the proliferation of BT-474_RICTOR$^{SH}$ cells (mean ± SEM, $n = 3$) compared to BT-474_SCRAM$^{SH}$ cells. (**D**) Tumor growth kinetics show significantly slower growth of BT-474–7_RICTOR$^{SH}$ (mean ± SEM, $n = 5$) tumors compared to BT-474_SCRAM$^{SH}$ tumors. (**E**) Altered glucosylceramide/ceramide ratio reveals a decrease in glucosylceramides in BT-474_RICTOR$^{SH}$ cells compared to BT-474 cells. (**F**, **G**) qRT-PCR (mean ± SEM, $n = 4$) (F) and immunoblots (G) demonstrate downregulation of UGCG without any change in GBA1 expression in BT-474_RICTOR$^{SH}$ cells compared to BT-474_SCRAM$^{SH}$ cells. (**H–J**) Absolute quantification (mean ± SEM, $n = 3–5$) of GM3 (H), GD3 (I), and GM2 (J) gangliosides shows a decrease in GM3 and GD3 gangliosides in BT-474_RICTOR$^{SH}$ cells compared to BT-474 cells. (**K**) Immunoblot shows the confirmation of UGCG overexpression in BT-474_RICTOR$^{SH}$ cells. (**L**) Fold changes in glucosylceramides (mean ± SEM, $n = 4$) in BT-474_ RICTOR$^{SH}$_UGCG$^{OE}$ cells compared to BT-474_RICTOR$^{SH}$ cells confirm an increase in glucosylceramides. (**M**) Cell proliferation assay demonstrates an increase in cell proliferation (mean ± SEM, $n = 3$) of BT-474_RICTOR$^{SH}$ cells on overexpression of UGCG. (**N–Q**) Immunoblot showing the confirmation of RICTOR knockdown in HCT-116_RICTOR$^{SH}$ (N) and HEK-293_RICTOR$^{SH}$ cells (P), and quantification of glucosylceramides in HCT-116_RICTOR$^{SH}$ (O) and HEK-293_ RICTOR$^{SH}$ (Q) cells compared to HCT-116 and HEK-293 cells. Data among groups were analyzed using an unpaired Student $t$ test or One-way ANOVA among multiple groups or Two-way ANOVA in time-dependent studies. *p*-value: \*\**p* < 0.01, \*\*\**p* < 0.001, \*\*\*\**p* < 0.0001. Numerical data can be found in S2 Dataset.
(S2_Fig.TIF)

**S3 Fig. Related to** Fig 3. (**A**) qRT-PCR (mean ± SEM, $n = 4$) confirms reduced expression of RICTOR-regulated ELF1, ZFX, and CTCF transcription factors in BT-474_RICTORSH cells. (**B**) University of California, Santa Cruz browser view of ZFX peaks on the promoter region of the UGCG gene as determined by ChIP-seq analysis. (**C**) ChIP-qPCR (mean ± SEM, $n = 3$) results show reduced binding of ZFX to the UGCG promoter in BT-474_RICTOR$^{SH}$ cells. (**D**) EMSA shows the binding of ZFX to the UGCG promoter (lanes 2 and 3) in BT-474–7_ZFX$^{OE}$ cells, shift-ablation assay in BT-474_ZFX$^{OE}$ cells (lane 4), competition assay with specific (lane 5) and unrelated oligo as a control (lane 6). (**E**) EMSA comparing endogenous ZFX-DNA binding activity between BT-474_SCRAM$^{SH}$ and BT-474_RICTOR$^{SH}$ cells. (**F**, **G**) Immunoblot and its quantification (mean ± SEM, $n = 3$) show downregulation of ZFX expression in BT-474_RICTOR$^{SH}$ cells. (**H**, **I**) Immunoblots and their quantification (mean ± SEM, $n = 3$) confirm that silencing of ZFX decreases UGCG expression (H) and overexpression of ZFX enhances UGCG expression in BT-474 cells (I). (**J**) The glucosylceramide to ceramide ratio confirms an increase in glucosylceramides in BT-474–7_ZFX$^{OE}$ cells. (**K–N**) Absolute quantification (mean ± SEM, $n = 3–4$) of GM3 (K), GD3 (L), GM2 (M), and GM1 (N) gangliosides shows an increase of GD3 and GM2 gangliosides and a decrease of GM3 and GM1 gangliosides in BT-474_ZFX$^{OE}$ cells. (**O**, **P**) Cell proliferation assay confirms an increase in cell proliferation of BT-474_ZFX$^{OE}$ cells (mean ± SEM, $n = 5$) (O), whereas BT-474_ZFX$^{SL}$ cells (mean ± SEM, $n = 3$) show decreased cell proliferation (P). (**Q**) Tumor growth kinetics recorded a significantly higher growth of BT-474_ZFX$^{OE}$ (mean ± SEM, $n = 5$) than

BT-474_VECT$^{OE}$ tumors. (**R, S**) Cell proliferation demonstrates a decrease in proliferation of BT-474_ZFX$^{OE}$ cells on UGCG silencing (mean±SEM, $n=4$) (R), whereas BT-474_ZFX$^{SH}$ cells (mean±SEM, $n=3$) show enhanced cell proliferation on UGCG overexpression (S). (**T**) Changes in gangliosides (normalized) (mean±SEM, $n=3$–5) on overexpression of UGCG in MCF-7_ZFX$^{SH}$ and BT-474_ZFX$^{SH}$ cells. (**U**) Immunoblots confirm that silencing of ZFX decreases UGCG expression in HCT-116, HEK-293, HepG2 and MDA-MB-453 cells. Data among two groups were analyzed using an unpaired Student $t$ test, among multiple groups using One-way ANOVA, and by Two-way ANOVA in time-dependent studies. $p$-value: *$p<0.$ , **$p<0.01$, ***$p<0.001$. Numerical data can be found in S3 Dataset.
(S3_Fig.TIF)

**S4 Fig. Related to** Fig 4. (**A**) Immunoblot showing alteration in phosphorylation of DNMT1 by pan-phospho-ser antibody in BT-474_RICTORSH and BT-474_ZFXOE cells compared to BT-474 cells. (**B**) Immunoblots showing a dose-dependent decrease in pAKT and UGCG expression in BT-474_ZFX$^{OE}$ cells on treatment with AKT inhibitor MK2206. (**C**) Results from qRT-PCR (mean±SEM, $n=4$) show a decrease in *UGCG* expression in BT-474_ZFX$^{OE}$ cells on AKT inhibition by MK2206 (4 µM). (**D**) Immunoblot reveals a decrease in phosphorylation of DNMT1 by pan-phospho-ser antibody on treatment of BT-474_ZFX$^{OE}$ cells with AKT inhibitor MK2206. (**E, F**) Results from qRT-PCR (mean±SEM, $n=4$) (E) and immunoblot (F) show increased *UGCG* expression in BT-474_RICTOR$^{SH}$ cells on DNMT inhibition by DAC. (**G**) ChIP-qPCR results (mean±SEM, $n=3$) confirm enhanced binding of ZFX to UGCG promoter in BT-474_RICTOR$^{SH}$ cells on DAC (5 µM) treatment. (**H, I**) Cell proliferation confirms increased proliferation of MCF-7_RICTOR$^{SH}$ (mean±SEM, $n=3$) (H) and BT-474_RICTOR$^{SH}$ (mean±SD, $n=3$) cells (I) on DAC treatment. (**J**) Immunoblot showing the change in phosphorylation of KDM5A using the pan-phospho-Ser antibody in BT-474_RICTOR$^{SH}$ cells compared to BT-474 cells. (**K**) Immunoblot shows alterations in the levels of KDM5A in nuclear and cytoplasmic extracts in BT-474_RICTOR$^{SH}$ cells compared to BT-474 cells. (**L**) Immunoblot reveals a decrease in phosphorylation of KDM5A on treatment of BT-474_ZFX$^{OE}$ cells with AKT inhibitor MK2206. (**M, N**) Results from qRT-PCR (mean±SEM, $n=4$) (M) and immunoblot (N) show increased *UGCG* expression in BT-474_RICTOR$^{SH}$ on KDM5A inhibition. (**O**) ChIP-qPCR results (mean±SEM, $n=3$) show a reduced H3K4Me3 mark on UGCG promoter in BT-474_RICTOR$^{SH}$ cells that increases on treatment with KDOAM-25 inhibitor (30 µM). (**P, Q**) Cell proliferation assay confirms increased proliferation of MCF-7_RICTOR$^{SH}$ (mean±SEM, $n=3$) (P) and BT-474_RICTOR$^{SH}$ cells (mean±SEM, $n=3$) (Q) on KDM5A inhibition. Data among the two groups were analyzed using an unpaired Student $t$ test, among multiple groups using One-way ANOVA, and by Two-way ANOVA in time-dependent studies. $p$-value: *$p<0.05$, **$p<0.01$, ***$p<0.0005$, ****$p<0.0001$. Numerical data can be found in S4 Dataset.
(S4_Fig.TIF)

**S5 Fig. Related to** Fig 5. (**A, B**) Immunoblots showing changes in the expression of different enzymes of the ganglioside metabolic pathway in MCF-7_RICTORSH, MCF-7_UGCGOE, and MCF-7_ZFXOE cells compared to MCF-7 cells (A) and in BT-474_RICTORSH, BT-474_UGCGOE, and BT-474_ZFXOE cells compared to BT-474 cells (B). (**C**) Immunoblots reveal elevated pEGFR$^{Y1173}$ and pEGFR$^{Y1068}$ levels in BT-474_UGCG$^{OE}$ and BT-474–7_ZFX$^{OE}$ cells compared to BT-474 cells. (**D**) Immunoblots show attenuated EGFR activation on shRNA-mediated silencing of GD3 synthase (ST8SIA1) in BT-474 cells. (**E**) Cell proliferation assay demonstrates a decrease in cell proliferation (mean±SEM, $n=3$) of BT-474_ST8SIA1$^{SH}$ cells compared to BT-474_SCRAM$^{SH}$ cells. (**F**) Immunoblots show enhanced EGFR activation on overexpression of ST8SIA1 in BT-474 cells. (**G**) Cell proliferation assay demonstrates increased proliferation (mean±SEM, $n=3$) of BT-474_ST8SIA1$^{OE}$ cells compared to BT-474_VECT$^{OE}$ cells. (**H**) Absolute quantification (mean±SEM, $n=3$) of GD3 gangliosides validates the silencing and overexpression of ST8SIA1 in BT-474 cells. (**I**) Immunoblots confirm overexpression of B3GALT4 in BT-474 cells. (**J**) Cell proliferation assay (mean±SEM, $n=4$) of BT-474_B3GALT4$^{OE}$ cells. (**K**) Absolute quantification (mean±SEM, $n=4$–5) of gangliosides validates the overexpression of B3GALT4 in BT-474 cells with an increase in GM1 gangliosides and a decrease in GD3 gangliosides. (**L**) Cell proliferation assay (mean±SEM, $n=3$) showing an increase in proliferation of BT-474_RICTOR$^{SH}$ cells upon supplementing GD3 gangliosides. (**M**) Immunoblots show attenuated EGFR activation on siRNA-mediated silencing

of ST8SIA1 in BT-474_ZFX$^{OE}$ cells. (**N**) Cell proliferation assay (mean±SEM, $n=3$) demonstrates a decrease in cell proliferation of BT-474_ZFX$^{OE}$ cells on siRNA-mediated inhibition of ST8SIA1. Data among two groups were analyzed using an unpaired Student $t$ test, among multiple groups using One-way ANOVA, and by Two-way ANOVA in time-dependent studies. $p$-value: *$p<0.05$, **$p<0.01$, ***$p<0.001$, ***$p<0.0001$. Numerical data can be found in S5 Dataset.
(S5_Fig.TIF)

**S6 Fig. Related to** Fig 6. (**A**) Schematic showing the PAM50 classification of the METABRIC tumor dataset used for analysis. (**B**, **C**) Gene expression of *UGCG* (B) and *ZFX* (C) in different breast cancer subtypes (PAM50) of the METABRIC dataset confirms high expression of *UGCG* and *ZFX* in luminal subtypes compared to other subtypes. (**D**–**G**) Change in expression of *UGCG* (D, F) and *ZFX* (E, G) with respect to ER (D, E) and PR (F, G) status in breast tumors of the METABRIC dataset confirms high *UGCG* (D, F) and high *ZFX* (E, G) expression in ER$^+$ and PR$^+$ tumors. (**H**) Percentage of tumors having high expression of *UGCG* and *ZFX* among luminal subtype tumors in the METABRIC data set. (**I**) UMAP analysis of a single-cell RNA sequencing dataset of breast cancer patients displayed nine distinct clusters of cells. (**J**) Dot plots represent the average expression (color) and the proportion of cells expressing *RICTOR*, *UGCG*, and *ZFX* across all cell clusters. (**K**) Changes (Mean±SEM, $n=3$–5) in gangliosides in eliglustat-treated tumors compared to untreated MCF-7 and BT-474 tumors. Data among two groups were analyzed using an unpaired Student $t$ test and among multiple groups using One-way ANOVA. $p$-value: *$p<0.05$, **$p<0.01$, ****$p<0.0001$. Numerical data can be found in S6 Dataset.
(S6_Fig.TIF)

**S1 Table. Clinical and pathological information of luminal breast cancer (ER+PR+HER2⁻) female patients of Indian origin included in this study.**
(S1_Table.DOCX)

**S2 Table. Immunohistochemistry data of tissue microarray from tumor samples of Indian female breast cancer patients showing the score for UGCG and ZFX staining.**
(S2_Table.DOCX)

**S3 Table. List of primers (human) used for validation of endogenous gene expression by real-time PCR.**
(S3_Table.DOCX)

**S4 Table. List of primers (human) targeting UGCG promoter used for validation of ChiP events.**
(S4_Table.DOCX)

**S5 Table. Table showing parameters for the estimation of gangliosides from cell lines.**
(S5_Table.DOCX)

**S1 Data. Numerical data related to** Fig 1.
(S1_Data.XLSX)

**S2 Data. Numerical data related to** Fig 2.
(S2_Data.XLSX)

**S3 Data. Numerical data related to** Fig 3.
(S3_Data.XLSX)

**S4 Data. Numerical data related to** Fig 4.
(S4_Data.XLSX)

**S5 Data. Numerical data related to** Fig 5.
(S5_Data.XLSX)

**S6 Data. Numerical data related to Fig 6.**
(S6_Data.XLSX)

**S1 Dataset  Numerical data related to S1 Fig.**
(S1_Data.XLSX)

**S2 Dataset  Numerical data related to S2 Fig.**
(S2_Data.XLSX)

**S3 Dataset  Numerical data related to S3 Fig.**
(S3_Data.XLSX)

**S4 Dataset  Numerical data related to S4 Fig.**
(S4_Data.XLSX)

**S5 Dataset  Numerical data related to S5 Fig.**
(S5_Data.XLSX)

**S6 Dataset  Numerical data related to S6 Fig.**
(S6_Data.XLSX)

**S1 Raw images.  Raw images of** Figs 1I, 1R–1S, 2A–2B, 2H, 2N, 3D–3F, 3H, 3J, 4B–4C, 4E, 4G, 4J–4K, 4M, 4O, 5A–5B, 5D, 5G, 5K, **S1C, S1J, S1L, S1Q, S2A–S2B, S2G, S2K, S2N, S2P, S3D–S3F, S3H–S3I, S3U, S4A–S4B, S4D, S4F, S4J–S4L, S4N, S5A–S5D, S5F, S5I,** and **S5M.**
(PDF)

## Acknowledgments

We thank Dr. Sagar Sengupta, Dr. Vinay Nandicoori, Dr. Kaustav Bandopadhyay, and Dr. Tapas Mukherjee for many helpful discussions. pERK1/2 antibody was a kind gift from Dr. Vinay Nandicoori. We thank DST-FIST sponsored Amity Lipidomics Research Facility at Amity University, Haryana. We thank the biorepository of Rajiv Gandhi Cancer Institute and Research Centre (RGCIRC), Delhi. We acknowledge NCBI (NIH), dbGAP for providing access to TCGA-BRCA cohort data. M.N.A., T.P., and A.K. thank ICMR, and S.J., D.J., and K. Rana thank CSIR for research fellowships. We are grateful to Nadathur Estates for their support of all the breast cancer research activities at SJRI. We acknowledge the support of the DBT e-Library Consortium (DeLCON) for providing access to e-resources. Research in the U.D. group is supported by ANRF, ICMR, and Koita Centre for Digital Health-Ashoka, Trivedi School of Biosciences, Ashoka University. Animal work in the small animal facility of the Regional Centre for Biotechnology is supported by Department of Biotechnology. AB thank the core funding from Regional Centre for Biotechnology. Fig 7 was drawn using BioRender.com with publication licence number DX28BR1W7V.

## Author contributions

**Conceptualization:** Avinash Bajaj, Ujjaini Dasgupta.

**Data curation:** Mohammad Nafees Ansari, Somesh K. Jha, Ali Khan, Kajal Rajput, Nishant Pandey, Dolly Jain, Rajeshwari Tripathi, Nihal Medatwal, Pankaj Sharma, Sudeshna Datta, Animesh Kar, Trishna Pani, Sk Asif Ali, Kaushavi Cholke, Kajal Rana, Valiya P. Snijesh, Jyoti S. Prabhu.

**Formal analysis:** Mohammad Nafees Ansari, Somesh K. Jha, Ali Khan, Kajal Rajput, Rajeshwari Tripathi, Nihal Medatwal, Sudeshna Datta.

**Funding acquisition:** Arnab Mukhopadhyay, Avinash Bajaj, Ujjaini Dasgupta.

**Investigation:** Mohammad Nafees Ansari, Somesh K. Jha, Ali Khan, Kajal Rajput, Nishant Pandey, Dolly Jain, Rajeshwari Tripathi, Nihal Medatwal, Pankaj Sharma, Sudeshna Datta, Animesh Kar, Trishna Pani, Sk Asif Ali, Kaushavi Cholke, Kajal Rana, Valiya P. Snijesh.

**Methodology:** Mohammad Nafees Ansari, Somesh K. Jha, Ali Khan, Kajal Rajput, Nishant Pandey, Nihal Medatwal, Sudeshna Datta, Animesh Kar, Sk Asif Ali, Valiya P. Snijesh.

**Project administration:** Avinash Bajaj, Ujjaini Dasgupta.

**Resources:** Suryanarayana V.S. Deo, Ashutosh Mishra, Jyothi S. Prabhu, Arnab Mukhopadhyay, Avinash Bajaj, Ujjaini Dasgupta.

**Supervision:** Soumen Basak, Arnab Mukhopadhyay, Avinash Bajaj, Ujjaini Dasgupta.

**Validation:** Mohammad Nafees Ansari, Somesh K. Jha, Ali Khan, Kajal Rajput.

**Visualization:** Mohammad Nafees Ansari, Somesh K. Jha, Ali Khan, Kajal Rajput, Avinash Bajaj, Ujjaini Dasgupta.

**Writing:** Mohammad Nafees Ansari, Somesh K. Jha, Avinash Bajaj, Ujjaini Dasgupta.

**Writing – review & editing:** Mohammad Nafees Ansari, Somesh K. Jha, Avinash Bajaj, Ujjaini Dasgupta.

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
