## [Editor Report · Decision Letter 0]

8 Apr 2024

Dear Dr Dasgupta,

Thank you for submitting your manuscript entitled "RICTOR Drives ZFX-mediated Ganglioside Biosynthesis to Promote Breast Cancer Progression" for consideration as a Research Article by PLOS Biology. Please accept my apologies for the delay in getting back to you as we consulted with an academic editor about your submission.

Your manuscript has now been evaluated by the PLOS Biology editorial staff, as well as by an academic editor with relevant expertise, and I am writing to let you know that we would like to send your submission out for external peer review.

Once your full submission is complete, your paper will undergo a series of checks in preparation for peer review. After your manuscript has passed the checks it will be sent out for review. To provide the metadata for your submission, please Login to Editorial Manager (https://www.editorialmanager.com/pbiology) within two working days, i.e. by Apr 10 2024 11:59PM.

Kind regards,

Richard

Richard Hodge, PhD

rhodge@plos.org

PLOS

---

## [Decision Letter · Decision Letter 1]

13 Jun 2024

Dear Dr Dasgupta,

Thank you for your patience while your manuscript "RICTOR Drives ZFX-mediated Ganglioside Biosynthesis to Promote Breast Cancer Progression" was peer-reviewed at PLOS Biology. Your manuscript has been evaluated by the PLOS Biology editors, an Academic Editor with relevant expertise, and by several independent reviewers.

As you will see in the reviewer reports, which can be found at the end of this email, although the reviewers find the work potentially interesting, they have also raised a substantial number of important concerns. Based on their specific comments and following discussion with the Academic Editor, it is clear that a substantial amount of work would be required to meet the criteria for publication in PLOS Biology. However, given our and the reviewer interest in your study, we would be open to inviting a comprehensive revision of the study that thoroughly addresses all the reviewers' comments. Given the extent of revision that would be needed, we cannot make a decision about publication until we have seen the revised manuscript and your response to the reviewers' comments. Your revised manuscript would need to be seen by the reviewers again, but please note that we would not engage them unless their main concerns have been addressed.

Having discussed the reviews with the Academic Editor, we think you should include the experiments that the reviewers ask for. Specifically, you should add additional experiments with a broader range of gangliosides, and validate your findings in additional cell lines.

We appreciate that these requests represent a great deal of extra work, and we are willing to relax our standard revision time to allow you 6 months to revise your study. Please email us (plosbiology@plos.org) if you have any questions or concerns, or envision needing a (short) extension.

**IMPORTANT - SUBMITTING YOUR REVISION**

*Resubmission Checklist*

*Published Peer Review*

*PLOS Data Policy*

*Blot and Gel Data Policy*

Sincerely,

Suzanne

Suzanne De Bruijn, PhD,

Associate Editor

PLOS Biology

sbruijn@plos.org

REVIEWS:

Reviewer #1: The manuscript by Mohd. Nafees, et al. titled "RICTOR Drives ZFX-mediated Ganglioside Biosynthesis to Promote Breast Cancer Progression" aims to connect MTORC2 signaling and epigenetic modulation of UGCG CpG islands through histone demethylation and DNMT1 activity as drivers of breast cancer tumor progression. The authors present a combination of sphingolipidomic results and protein expression in a myriad of OE and shRNA modulated biological systems within the MTORC2 to UGCG axis, in both in vitro and in vivo settings. They observe that over-expression of RICTOR, ZFX, and UGCG increases the ratio of the respective glucosylceramides to ceramides while silencing of genes in this axis reduce cell proliferation, tumor growth, and migration. Then, they altered the epigenetic landscape in vitro to connect phosphorylation downstream of RICTOR to reduce CpG methylation by DNMT1 enabling ZFX transcription of UGCG. Lastly, upon establishing the RICTOR-ZFX-UGCG connection, they perform targeted lipidomic analysis to characterize how altered GSL species in the UGCG, RICTOR, and ZFX overexpression systems yield different GSL profiles that affect EGFR signaling and cell proliferation. They conclude by comparing TCGA datasets and clinical samples from an Indian cohort with their in vitro and in vivo results, highlighting the prevalence of UGCG and ZFX expression in luminal breast cancer and that inhibition by eliglustat reduces growth for two luminal cell lines in NOD mice. Therefore, the manuscript provides a logical progression through analyzing the link between RICTOR and glycosphingolipid profiles that supports luminal breast cancer progression.

The manuscript evaluates how modulating RICTOR and its downstream substrates influence UGCG expression and GSL profiles and abundances using a myriad of cell biology approaches in both in vitro and in vivo models. The stepwise progression through the RICTOR-pAKT-ZFX-UGCG axis they present as supportive for tumor growth is compelling, but some revisions and clarifications would help to answer some remaining questions.

Major points:

1. One of the surprising findings is how overexpression of UGCG and ZFX result in a decrease of GM1. The authors show no difference in the enzymes for GSL synthesis, but do not explore whether there are changes in enzymes for breaking down complex GSL. Evaluating expression of HEXA/B, GLA/GLB would help elucidate this difference. Additionally, showing the changes to the GSL profile upon ST8SIA1siRNA treatment would provide a more complete picture.

2. The authors do not show GM2/GD2 profiles throughout the paper; however these have also been shown to promote tumor growth and RTK signaling. While the impact of UGCG OE and increased GSLs to luminal breast cancer progression is clear, to show mechanistic impact of GSLs as the authors claim, it would be beneficial to include all major gangliosides in the pathway.

3. The experiment performing the addition of GM1 and GD3 is not clear. In the methods, the authors describe dissolving both GSLs in serum free media but not how the cells were cultured. If the experiments were performed in serum free media, the elimination of growth factors and other components present in serum make it difficult to establish the impact solely of extracellular GSLs. While adding exogenous GSLs is not the same as those properly trafficked and synthesized endogenously, the authors should at least use delipidated FBS to maintain serum components. If serum was used in the media, methods need to be revised. Using an OE system for ST8SIA1 and B3GALT4 would be a cleaner way to increase flux towards those GSLs and capture the impact of endogenous synthesis

Minor points:

* The authors use the ratio of glucosylceramides to their respective ceramides to show that there is greater UGCG expression. It would be good to demonstrate that GSLs downstream of glucosylceramide are also increased, since the GSLs with greatest impact on signaling and RTK activity are downstream of UGCG activity.

* In Figure 2O, the authors should include the lipidomic profile or comparison to show if UGCG OE with ZFXSL impacts the GSLs in the cell even with a rescue.

* In figure 3R, the authors show how ZFXOE +UGCGSL exhibits reduced growth as compared to just ZFXOE, but a comparison with a UGCGSL would benefit to deconvolute the impact of ZFXOE to other binding sites that may promote growth.

* Some of the fold changes presented don't seem to match those of the figures. Fig 5B for example does not match the 4-fold described in the text.

* The authors focus on ganglioside metabolism, but neutral glycosphingolipids will also be altered because of UGCG overexpression. Showing how neutral GSLs that will impact membrane dynamics and cell-cell contacts will aid in generating a mechanistic picture.

* One of the limitations in performing the experiments in NOD mice is losing the immune interactions that will be altered as cells present an increase of GSLs, altering their interactions with other tumor cells, extracellular matrix, and the immune system. This should be discussed as a limitation, or some work could be done to elucidate how changes in GSL profile affect growth.

* No methods are shown for BT474 flank injections.

* A discussion about why cells were not injected into mammary fat pads would be good as this would be more representative of the local environment.

* The authors focused on luminal breast cancer lines due to TCGA dataset results. However, they also tested siRNA on HCT116s, HEPG2s, and MDA-MB-453s (TNBC) and saw a similar reduction in UGCG expression upon ZFXSL. Some discussion or results suggesting whether this response of increased growth because of RICTOR-ZFX-UGCG axis is only present in luminal breast cancer, or other lines would help to highlight the impact of these results for combination therapies and strategies as the authors mention in the discussion.

Reviewer #2: In the current study Mohd. Nafees Ansari et al investigated the regulatory mechanisms controlling sphingolipid and ganglioside biosynthesis in mammalian cells, specifically in luminal breast cancer cells. They concluded that RICTOR, as part of mTORC2, regulates the synthesis of these lipids through transcriptional and epigenetic mechanisms, involving the modulation of UGCG expression by ZFX and AKT substrates, thereby promoting tumour growth and activating the EGFR signaling pathway.

The topic is very interesting. Glycolipid metabolism has established roles in cancer biology and therapy response, yet the field has lacked substantial efforts of mapping the regulation of these metabolic and signaling pathways. Although of interest, the study suffers from a number of shortcomings, including the limited range of cell lines used, incomplete confirmation of key experiments, methodological issues in the in vivo experiments, and importantly, the substantial lack of supplementation experiments with glucosylceramide and complex glycolipids, which limit the ability to draw solid conclusions on the molecular mechanisms explored.

Major comments

1. The observation that RICTOR regulates glycolipid metabolism is interesting and important. However, authors show this observation only in 2 breast cancer cell lines. It would be interesting to at least demonstrate this key finding in a broader range of cells (including non-diseased cells).

2. Moreover, authors use the BT-474 cell line to confirm their findings in MCF-7 cells, but only include selected experiments and do not confirm multiple key experiments in the second cell line.

3. Regarding the in vivo experiment with Eliglustat: the method of administration is not an accepted method. Also the single dose is low (25 mg/kg), and lacks biomarker confirmation. Moreover, the reference (ref 27) given to the details of the method does not contain further details of the method. Finally, the control arm for this experiment was "untreated" animals - lacking vehicle control. Taken together, these points severely limit any conclusion from Fig. 6L-O.

4. The major limitation of this study is the substantial lack of experiments with supplementation of glucosylceramide as well as more complex glycolipids such as gangliosides. The authors only include 1 example of such phenocopy/rescue experiment where GD3 or GM1 ganglioside is supplemented to RICTOR KD or UGCG OE MCF-7 cells, respectively (Fig. 5N,O). However, the use of untreated control instead of vehicle control, as well as the double normalization (untreated condition for all cells set to 100%) limits conclusive interpretation of these data. In general, the absence of such data prevents any solid conclusion on the molecular mechanism that the authors wanted to explore in this manuscript.

Minor comments

1. The authors show elevated expression of RICTOR, ZFX, and UGCG in bulk RNA data from patients with cancer to suggest elevated expression in cancer cells in situ. However, it is important to verify cancer cell specific overexpression by analysis of publicly available or proprietary single cell RNA data.

2. Authors use UGCGDEAD as a control in several experiments, but fail to include a confirmation experiment that demonstrates the inactive state of this gene product they overexpress. This could be done by lipidomics.

3. Lipidomics data from patients where selected ceramide and glucosylceramide species are compared between tumor and adjacent normal tissues are now represented and statistically analyzed in an unpaired manner. However, given these samples are "matched" (from the same patients), they should be analyzed as such with a paired statistical test and this could have major implications on the observed effect and associated conclusion.

---

## [Decision Letter · Decision Letter 2]

3 Jul 2025

Dear Dr Dasgupta,

Thank you for your patience while we considered your revised manuscript "RICTOR Drives ZFX-mediated Ganglioside Biosynthesis to Promote Breast Cancer Progression" for publication as a Research Article at PLOS Biology. This revised version of your manuscript has been evaluated by the PLOS Biology editors, the Academic Editor and the original reviewers.

Based on the reviews, I am pleased to say that we are likely to accept this manuscript for publication, provided you satisfactorily address the remaining points raised by Reviewer #2. Please also make sure to address the following data and other policy-related requests that I have provided below (A-J):

(A) We routinely suggest changes to titles to ensure maximum accessibility for a broad, non-specialist readership. In this case, we would suggest a minor edit to the title, as follows. Please ensure you change both the manuscript file and the online submission system, as they need to match for final acceptance:

"The mTORC2 subunit RICTOR drives breast cancer progression by promoting ganglioside biosynthesis through transcriptional and epigenetic mechanisms”

(B) In the ethics statement in the Methods section, please include the full name of the Institutional Review Board/human ethics committee that reviewed and approved the use of patient tumour samples in the study. In addition, please include the specific approval number that you obtained from the IRB/ethics committee.

(C) Please include information about the form of consent (written/oral) given for research involving human participants.

(D) In the animal ethics statement in the Methods section, please include the full name of the IACUC/ethics committee that reviewed and approved the animal care and use protocol/permit/project license. Please also include the specific approval number.

(E) You may be aware of the PLOS Data Policy, which requires that all data be made available without restriction: http://journals.plos.org/plosbiology/s/data-availability. For more information, please also see this editorial: http://dx.doi.org/10.1371/journal.pbio.1001797

-Supplementary files (e.g., excel). Please ensure that all data files are uploaded as 'Supporting Information' and are invariably referred to (in the manuscript, figure legends, and the Description field when uploading your files) using the following format verbatim: S1 Data, S2 Data, etc. Multiple panels of a single or even several figures can be included as multiple sheets in one excel file that is saved using exactly the following convention: S1_Data.xlsx (using an underscore).

-Deposition in a publicly available repository. Please also provide the accession code or a reviewer link so that we may view your data before publication.

Figure 1B-H, 1J-Q, 2C-G, 2H-M, 2O-P, 3B-C, 3G, 3I, 3K-W, 4D, 4F, 4H, 4L, 4N, 4P, 5C, 5E-F, 5H-J, 5L-M, 6B-H, 6J-O, S1A-B, S1D-I, S1K, S1M-P, S2C-F, S2H-J, S2L-M, S2O, S2Q, S3A-C, S3G-T, S4C, S4E, S4G-I, S4M, S4O-Q, S5E, S5G-H, S5J-L, S5N, S6B-K

(F) Please also ensure that each of the relevant figure legends in your manuscript include information on *WHERE THE UNDERLYING DATA CAN BE FOUND*, and ensure your supplemental data file/s has a legend.

(G) We require the original, uncropped and minimally adjusted images supporting all blot and gel results reported in the following Figures:

Figure 1I, 1R-S, 2A-B, 2H, 2N, 3D-F, 3H, 3J, 4B-C, 4E, 4G, 4J-K, 4M, 4O, 5A-B, 5D, 5G, 5K, S1C, S1J, S1L, S1Q, S2A-B, S2G, S2K, S2N, S2P, S3D-F, S3H-I, S3U, S4A-B, S4D, S4F, S4J-L, S4N, S5A-D, S5F, S5I, S5M

We will require these files before a manuscript can be accepted so please prepare and upload them now. Please carefully read our guidelines for how to prepare and upload this data: https://journals.plos.org/plosbiology/s/figures#loc-blot-and-gel-reporting-requirements

(H) Per journal policy, if you have generated any custom code during the course of this investigation, please make it available without restrictions. Please ensure that the code is sufficiently well documented and reusable, and that your Data Statement in the Editorial Manager submission system accurately describes where your code can be found.

(I) Please note that per journal policy, the model system/species studied should be clearly stated in the abstract of your manuscript (e.g. human and mouse).

(J) Please ensure that your Data Statement in the submission system accurately describes where your data can be found and is in final format, as it will be published as written there.

We expect to receive your revised manuscript within two weeks.

*Published Peer Review History*

*Press*

Best regards,

Richard

Richard Hodge, PhD

rhodge@plos.org

Reviewer remarks:

Reviewer #1: The authors have addressed all of my concerns.

Reviewer #2: The authors have performed a substantial amount of additional work that has significantly improved the quality of the manuscript.

Major comment: Figure 6O shows increased abundance of glucosylceramides in BT-474 tumors upon Eliglustat treatment, in contrast to Figure 6L and expected effects of UGCGi. The manuscript currently lacks in a description and a discussion for this finding.

Minor comments: Overall, the manuscript would benefit from more precise phrasing when reporting increases or decreases, as it often omits the specific baseline or comparator to which these changes refer. It also lacks essential details on the timing of experiments, such as the duration of cell or mouse treatments prior to measurements, which is critical for reproducibility of results.

---

## [Editor Report · Decision Letter 3]

12 Aug 2025

Dear Ujjaini,

On behalf of my colleagues and the Academic Editor, Heather Christofk, I am pleased to say that we can accept your manuscript for publication, provided you address any remaining formatting and reporting issues. These will be detailed in an email you should receive within 2-3 business days from our colleagues in the journal operations team; no action is required from you until then. Please note that we will not be able to formally accept your manuscript and schedule it for publication until you have completed any requested changes.

PRESS

Best wishes,

Richard 

Richard Hodge, PhD

rhodge@plos.org

PLOS
